# The five main influencing factors on lidar errors in complex terrain

Tobias Klaas-Witt[1], Stefan Emeis[2, 3]

[1]Fraunhofer Institute for Energy Economics and Energy System Technology (IEE), Germany
[2]Institute of Geophysics and Meteorology, University of Cologne, Germany
[3]Institute of Meteorology and Climate Research, Karlsruhe Institute of Technology, Germany

*Correspondence to*: Tobias Klaas (tobias.klaas-witt@iee.fraunhofer.de)

**Abstract.**

Lidars have become a valuable technology to assess the wind resource at hub height of modern wind turbines. However, because of the assumption of homogeneous flow in their wind vector reconstruction algorithms, common wind profile Doppler

lidars suffer from errors at complex terrain sites.

This study analyses the impact of the five main influencing factors on lidar measurement errors in complex terrain, i.e. orographic complexity, measurement height, surface roughness and forest, atmospheric stability and half-cone opening angle, in a non-dimensional, model-based parameter study.

In a novel approach, the lidar error $\varepsilon$ is split up into a part $\varepsilon_c$, caused by flow curvature at the measurement points of the lidar

and a part $\varepsilon_s$, caused by the local speed-up effects between the measurement points. This approach allows for a systematic and complete interpretation of the influence of the half-cone opening angle $\varphi$ of the lidar on the total lidar error $\varepsilon$. It also provides information about the uncertainty of simple lidar error estimations that are based on inflow and outflow angles at the measurement points.

The model-based parameter study is limited to two-dimensional Gaussian hills with hill height $H$ and hill half-width $L$. $H/L$

and $z/L$, with z being the measurement height, are identified as the main scaling factors for the lidar error. Three flow models of different complexity are used to estimate the lidar errors. The outcome of the study provides various findings that enable an assessment of the applicability of these flow models.

The study clearly shows that orographic complexity, roughness and forest characteristics, as well as atmospheric stability, have a significant influence on lidar error estimation. Based on the error separation approach it furthermore allows for an in-depth

analysis of the influence of reduced half-cone opening angles, explaining contradiction in the previously available literature.

The choice and parameterization of flow models and the design of methods for lidar error estimation are found to be essential to achieve accurate results. The use of a RANS CFD model in conjunction with an appropriate forest model is highly recommended for lidar error estimations in complex terrain, since forest (and roughness) tend to reduce the lidar error. If atmospheric stability variation at a measurement site plays a vital role, it should also be considered in the modelling. When

planning a measurement campaign, an accurate estimation of the predicted lidar error should be carried out in advance to choose a reasonable measurement location. This will decrease measurement uncertainties and maximize the value of the measurement data.

## 1 Introduction

Vertical profiling monostatic Doppler lidars are by far the most often used lidar systems in wind energy applications because they are readily available, flexible and easy to use even at remote locations (Gottschall et al., 2011; Klaas et al., 2015; Clifton et al., 2018).

Due to the assumption of homogeneous flow in their wind vector reconstruction algorithms, common Doppler lidar profilers suffer from erroneous reconstruction of the horizontal wind vector in inhomogeneous flow conditions that are usually found at complex terrain sites (Bradley et al., 2015). However, many wind farms are nowadays planned at such sites (Callies, 2014). It is, therefore, necessary to examine the accuracy of lidars at such sites, identify challenges, and find solutions in order to be able to keep measurement uncertainty on acceptable levels. The most promising approach to account for their lack of accuracy at complex terrain sites is the application of wind flow models (Bradley et al., 2015). Different studies on this approach are considered in the following literature overview that is a summary of the literature study carried out within the dissertation of the first author (Klaas, 2020). The literature review aims at identifying the most important influencing factors on the lidar error, which are then concluded at the end of the introductory section.

In order to be able to reconstruct the horizontal wind speed, Doppler lidars measure the radial velocity at different locations in the atmosphere. Under the assumption of identical wind speed at these locations (so-called "homogeneous flow assumption"), simple trigonometric functions can be used to calculate the horizontal wind speed. This assumption is not valid for measurement sites with significant spatial changes in wind speed, e.g. complex terrain sites (Courtney et al., 2008; Clive, Peter J. M., 2008).

One of the first relevant comparisons between a lidar and a 100 m mast was carried out by Antoniou et al. (2007). A ZephIR lidar was placed at a complex terrain test station in Greece. While showing correlation comparable to flat terrain studies on the one hand (compare e.g. Smith et al., 2006), a significant underestimation of the wind speed by the lidar was observed on the other hand. Courtney et al. (2008) stated that errors in the determination of mean wind speed in the order of 5-10 % are not uncommon for complex terrain sites. As a solution, they proposed to reduce the lidar cone angle from 30° to 15° in order to reduce the distance between the measurement points and consequently the magnitude of the difference in wind speed at the measurement points.

A first attempt to explain and model the error of monostatic remote sensing instruments in complex terrain (lidar and sodar) is presented by Bradley (2008). He applies a simple two-dimensional potential flow model to estimate the wind flow and uses the model results to correct the error due to the homogeneous flow assumption. Depending on the shape of the hill and the measurement height, he finds sodar errors between 5 and 20 % for a cone angle of 20°. Contrary to his hypothesis (and the one from Courtney et al. (2008)), there is no significant increase in the magnitude of the errors when increasing the cone angle to 30° (which is typical for most lidars).

Bingöl et al. (2009) study the lidar error at two complex terrain sites in Greece. Here the lidar errors of a ZephIR reach up to a magnitude of 10 % depending on wind direction and for heights between 30 and 80 m. An algorithm is implemented into WAsP Engineering (WEng) that uses the model results to calculate an estimation for the lidar error at the two sites. Although the model is simple and limited to low slopes only, the estimated errors fit well to the observed results for the main wind directions. For the wind directions with increased slope (south-west), the results do not fit very well, which is attributed to

limitations in WEng. In a simple, two-dimensional analytical consideration, Bingöl et al. (2009) also show that the lidar error is not dependent on the cone angle, but only on the horizontal homogeneity of the flow. These analytical findings are verified by Foussekis (2009), who compares the results for the ZephIR and two Windcube lidars, one with a 30° and one with a 15° cone angle against a 100 m mast at the complex terrain CRES test station in Greece. Here, an underestimation of about 6 % is found for all three lidars, independent of measurement principle (CW and pulsed) and cone angle (15° or 30°). Additionally,

the study states that the lidar error is not dependent on height at this site (Foussekis, 2009).

For the ZephIR and the Leosphere Windcube lidar, lidar error estimation methods using the Reynolds-averaged Navier-Stokes (RANS) computational fluid dynamics (CFD) models WindSim and Meteodyn WT were developed and tested over the years (Harris et al., 2010; Meissner and Boquet, 2011; Bezault et al., 2012; Jokela et al., 2013; Kim and Meissner, 2017). The lidar error estimation approaches are comparable to that proposed by Bingöl et al. (2009). The results emphasize that WEng tends

to overestimate the lidar error, especially for steep slopes. The CFD code provides a better estimate for the inhomogeneous flow above the terrain and corrected lidar data mostly shows a better agreement to the reference data after application of the CFD correction. The authors conclude that these findings show the limitations of the WEng model in terms of terrain complexity (compare e.g. Harris et al., 2010).

Based on his previous study, Bradley (2012) extends his potential flow model to sodar and lidar data on a simple two-

dimensional hill and an escarpment. Additionally, the results from the simple model are compared to more advanced RANS CFD models (WindSim and OpenFOAM) from Behrens et al. (2012) at two different complex terrain sites in Scotland and New Zealand. Results show that the simple potential flow model is mostly sufficient to estimate the lidar error at these sites (Bradley, 2012), although there are some cases where RANS CFD provides better results (Behrens et al., 2012).

For the Windcube v2, Leosphere introduced a proprietary method called "flow complexity recognition" (FCR) to correct for

the lidar error in complex terrain already during the measurement. With FCR the bias between lidar and mast is significantly reduced in the study of Foussekis (2011). FCR was also tested by Wagner and Bejdic (2014) at a complex terrain site. Here, wind speeds are underestimated by 4 % in default mode. With FCR turned on, there is a slight overestimation of 1.5 %. Although this might probably be within the measurement uncertainty, FCR tends to over-correct for the lidar error at this site. According to the authors, this has also been observed in previous studies (Wagner and Bejdic, 2014). In an inter-comparison

of FCR results for sites of various complexities, this effect is also evident for several sites, although the magnitude of the bias is mostly reduced by FCR. It is interesting to note that in this study, FCR heavily over-corrects the measurement data at the highly-complex site by 20 %, which implicates that FCR might not be suitable for this site (Krishnamurthy and Boquet, 2014). In 2017, Leosphere revealed the method behind FCR in a detailed technical report (Leosphere, 2017). It uses the 3D wind field

model "SWIFT" to calculate the wind flow in the closer proximity of the lidar to estimate the lidar error. It is, therefore, comparable to the other model-based correction approaches. Leosphere also states that FCR is limited to moderately complex terrain and low surface roughness (Leosphere, 2017). In its complexity, the model is comparable to WEng, which might explain its incapability to be applied at very complex and forested sites.

A systematic review over the at that time available studies and open research questions regarding remote sensing in complex terrain is given in Bradley et al. (2015). The study concludes that simple models, e.g., potential flow models, can often correct for the lidar error acceptably well. However, as soon as recirculation or detached flow situations occur, more sophisticated models are needed that are capable of modelling those flow features. Also, more detailed characteristics of the atmospheric boundary layer flow (e.g., low-level jets or atmospheric stability) are so far not treated in the context of remote sensing in complex terrain (Bradley et al., 2015).

Based on this literature review, it becomes obvious that the performance of lidar error estimation approaches based on flow modelling is heavily dependent on the actual site characteristics. Five governing influencing factors on the lidar error in complex terrain must be considered: Orographic complexity, terrain roughness and vegetation, atmospheric stability, measurement height and half-cone opening angle.

In all available studies on lidar-mast comparisons at complex terrain sites, it is found that the lidar error is dependent on orographic complexity. Lidar errors measured at sites of different complexity and for distinct wind directions vary in magnitude and can either be negative or positive (e.g Antoniou et al., 2007; Bingöl, 2009). The respective literature lacks a systematic comparison of lidar measurement accuracy concerning different orographic complexities. Existing experimental studies mostly focus on the results from a single site. Comparing different studies with sites of different orographic complexity is difficult, as the used anemometry and equipment, as well as the methods and definitions for data preparation and analysis, are usually not the same. In addition to that, it is not always the same type or even technology of wind lidar that is used for the evaluation, and the results (e.g., from pulsed and continuous-wave lidars) are not directly comparable.

Measurement sites do not only differ in terms of orographic complexity but also in land cover and, therefore, terrain roughness and vegetation. Many complex terrain sites that are used for current wind energy projects are located in forested terrain (Callies, 2014). It is well known that terrain roughness and especially forest heavily influence the wind flow above the terrain (Finnigan and Belcher, 2004; Belcher et al., 2008). Roughness elements and forest induce turbulence and shear in the wind profile or even enhance the formation of flow separation zones (Belcher et al., 2012; Shannak et al., 2012). With this in mind, the assessment of the individual influence of single parameters of different sites on the actual lidar error is challenging. Comparing the accuracy of lidar measurements between sites of different orographic complexity, for example, is hindered by the influence of terrain roughness and vegetation. For example, results from Klaas et al. (2015) at a forested site show much smaller lidar errors than those found in Bingöl et al. (2009) or Foussekis (2011), which are both not forested.

Nearly all studies have in common that the reference masts are lower than or equal to 100 m, as these are common and economically feasible mast heights. Contrary to that, wind turbines have been increasing in both hub height and rotor diameter, leading to upper tip heights of modern wind turbines in the range of 200 m (Rohrig, 2018).

Another aspect that has not yet been treated in literature is the influence of atmospheric stability on lidar measurement
accuracy. Although it is stated in a few studies that there might be an effect from this (e.g., Bradley et al. (2015)), the author is not aware of any piece of work that examines the dependence of lidar measurement accuracy in terms of varying atmospheric stability in any kind. However, there is a significant influence of atmospheric stability on the wind profile and the wind flow patterns over or around hilly terrain (Ross et al., 2004; Leo et al., 2016), that might very well influence lidar measurement accuracy. Atmospheric stability does have an influence on both: Speed-up effects and flow curvature which are the main
reasons for lidar errors in complex terrain (Ross et al., 2004; Emeis, 2018).

The half-cone opening angle of the lidar is another factor that must be considered in lidar error estimation. With increasing measurement height, the distance between the measurement points of the lidar also increases significantly. Courtney et al. (2008) were proposing to reduce the half-cone opening angle from 30° to 15° in order to reduce the lidar error in complex terrain. This suggestion is interrogated and tested experimentally by Bingöl et al. (2009) and Foussekis (2009), who come to
the conclusion that the half-cone angle does not influence the lidar error. Also, Bradley et al. (2015) derive lidar error estimations that are solely dependent on flow curvature and independent of the half-cone opening angle. However, these findings are based on the assumption of symmetric flow and constant flow curvature. For flow simulations that consider surface roughness and forest as well as atmospheric stability, these assumptions are not necessarily valid (compare e.g. Ross et al. (2004) and Belcher et al. (2008)).


In all studies mentioned above, mast-based cup anemometry is used as a reference to quantify the lidar errors in complex terrain and to evaluate the correction methods. In this context it is important to consider that the uncertainty of cup anemometers is significantly higher at complex terrain sites than in flat terrain (Dahlberg et al., 2006). Anemometer classifications according to IEC 61400-12-1 should be applied to assess the total uncertainty of cup anemometers in comparison to that from lidar
profilers (International Electrotechnical Commission, 2017). Classification numbers for flat and complex terrain for common cup anemometers have e.g. been derived in Pedersen and Busche (2006). They range from 1.5 to 1.8 in Class A and 2.9 to 3.8 in Class B for a Thies First Class cup anemometer, depending on the methodology and cup anemometer models used. For flat terrain, a class number of 1.5A results in a standard operational uncertainty of 0.87 % and for complex terrain, the class number of 2.9B results in an uncertainty of 1.6 %. Including a calibration uncertainty of 1.0 % and a mounting uncertainty of 0.5 %
leads to a standard uncertainty of 1.42 % in flat terrain and 1.95 % in complex terrain (International Electrotechnical Commission, 2017). Using the extended uncertainty (95 % confidence interval) doubles these uncertainties.

This study firstly analyses the influence of the most governing factors on lidar errors in complex terrain in a systematic way revealing the actual influence and importance of each. Secondly, it combines these findings to an overall perspective that can

be used as a practical guideline for the application of lidars in the terrain of various complexities. To the knowledge of the author, there is no comprehensive assessment like this so far. The findings of Klaas et al. (2015) were a trigger to intensify research on model parameterization in the context of lidar error estimation, which is done in the present study.

Moreover, due to the model-based approach of the study, it can also answer questions of the applicability and limitations as well as the strengths and weaknesses of the different flow models that are used to estimate the lidar error. It, therefore, provides helpful guidance on the necessary complexity of the flow model to be used for lidar error estimation in a particular situation or at a specific site.

Especially the work on atmospheric stability and its impact on the lidar error in complex terrain has not been treated in literature so far. The study therefore closes a gap between the knowledge about the influence of atmospheric stability from a
meteorological perspective on the one hand and its implications on the lidar error on the other hand.

Furthermore, the lidar error estimation follows a novel approach where the lidar error is separated into its two main parts: Lidar error due to flow curvature effects and lidar error due to speed-up effects. A comparable approach was not found in the relevant literature. This approach gives a more detailed and structured insight into the flow effects that cause lidar errors in
complex terrain. Especially the question, if the lidar half-cone opening angle is an important parameter that has to be considered, can be answered by this approach.

## 2 Methods

### 2.1.    General considerations

Using computational flow models, e.g. for wind resource assessments, presumes knowledge about the optimal parameterization
to fit the model to the considered site and measured wind profiles. From that, model results can be used for vertical and horizontal extrapolation of the wind conditions (Ayotte, 2008). From geodata, maps and site visits, information about land-use is gathered and transferred into model parameters such as roughness length and forest model parametrizations.

Wherever possible, modelled wind profiles are compared to those from measurements available at the site in order to improve
the accuracy of the model results (e.g. Palma et al., 2008). It is necessary to estimate the overall uncertainty in the measurement data, including additional uncertainties due to complex terrain, when comparing it to the model. If only lidar measurements are available at a complex terrain site, the measured data is additionally affected by the complex terrain lidar error. Because of that, additional uncertainties occur when using this data for model validation. Alternatively, a correction of the lidar data can be carried out, but then the additional uncertainty of this correction must be considered as well. However, when the lidar error
is significant, and, e.g., height dependent, it might be necessary to use other data sources for model validation in preparation of the lidar error correction. (FGW e.V., 2017; Clifton et al., 2018).

For the latter, an evaluation of uncertainties of currently used correction methods is necessary. Recent studies only state the "best" results that can be achieved by using a parameter set optimized for the actual measurement site. In order to do so, the model parameterization has either been validated against a measured wind profile from a close-by mast or the accuracy of the lidar error correction method itself is validated by a mast-lidar comparison at the site (e.g., Bingöl et al., 2009; Klaas et al., 2015). To the knowledge of the author, there is no systematic evaluation available that analyses the influence of model parameterization on the results of lidar error corrections.

## 2.2. Flow modelling methods

In this study, three different common flow models are used. Different parameter settings are applied to accomplish a systematic understanding of flow model parameter variations on the resulting lidar error estimations. Wind flow is modelled above simplified, two-dimensional Gaussian hills as e.g. used in Feng and Shen (2014) with hill height $H$ and hill half-width $L$ (see Figure 1 and Table 1):

$$z = H * \exp\left(-\frac{x^2}{L^2} log(2)\right) \tag{1}$$

Using Gaussian or Cosine hills as a basis for simplified terrain models is well established in literature. A recent review on this topic can be found in Finnigan et al. (2020). Results are presented in a non-dimensional way wherever possible. Under consideration of the measurement height $z$, two non-dimensional parameters are derived that allow for a well-arranged presentation of the results: $H/L$ as a parameter that expresses orographic complexity and $z/L$ that relates the measurement height to the actual size of the hills. Further parameters that are analysed are terrain roughness $z_0$, forest height $h$ and forest density, atmospheric stability and half-cone opening angle $\varphi$ of the lidar. The lidar is placed on top of the hill in this study.

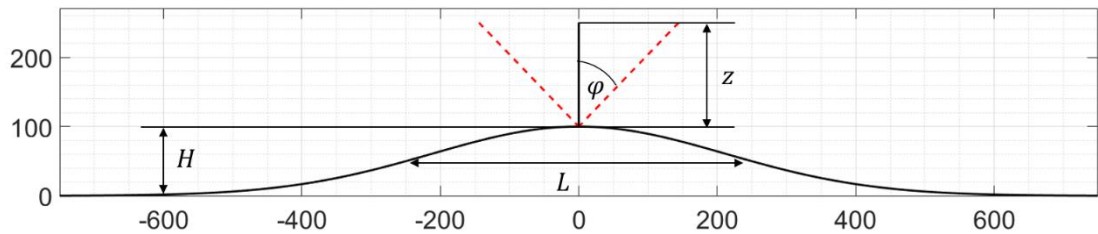

**Figure 1: Exemplary sketch of a Gaussian hill used in the parameter study. Each hill is defined by its height $H$ and its hill half-width $L$. A representation of the lidar geometry for measurement height $z$ and a half-cone opening angle $\varphi$ is shown as well. In this example, $H$ is 100 m and $L$ is 250 m resulting in an $H/L$ ratio of 0.4 and for the measurement height $z$ of 150 m in a $z/L$ ratio of 0.60.**

**Table 1: Set of Gaussian hill geometries used in the parameter study. The table provides information about the hill height $H$, the hill half-width $L$ and the corresponding ratio $H/L$. Addtionally the maximum slope (maximum terrain inclination) at the flanks of the hill is calculated for the four different used ratios.**

| H/L | L [m] | 50 | 100 | 150 | 200 | 250 | 500 | 750 | Max. slope |
|-----|-------|-----|------|------|------|------|------|------|------------|
| 0.1 |       | 5  | 10  | 15  | 20  | 25  | 50  | 75  | 0.07 |
| 0.2 | H [m] | 10 | 20  | 30  | 40  | 50  | 100 | 150 | 0.14 |
| 0.3 |       | 15 | 30  | 45  | 60  | 75  | 150 | 225 | 0.21 |
| 0.4 |       | 20 | 40  | 60  | 80  | 100 | 200 | 300 | 0.29 |

Three different steady-state flow models are used that differ in terms of complexity. This enables to analyse the influence of additional parameters and model extensions (e.g. a forest model) in reference to the simpler approaches.

First, the simple, two-dimensional potential flow model from Bradley (2008) is used to analyse the terrain effects on the wind flow. The model is frictionless and symmetric and does therefore not cover any effects of roughness, forest or other more complex properties of the atmospheric wind flow. It was run with a constant resolution of 5 m in the vertical and 10 m in the horizontal direction. It is used as a baseline case to depict the influence of the diverse parameters on the results of the other, more complex models. The model is implemented and adapted to the correction approach described below.

The second model that is used, WEng, is based on linearized Navier-Stokes Equations and a very common model for wind energy applications (Mann et al., 2002). It is able to model the influence of roughness on the wind flow. However, with regards to the available literature, the model is recommended to be used only for slightly complex terrain and there is no forest model implemented (Mann et al., 2002; Dellwik et al., 2006). A displacement height is not used within this study. The model is also used by Bingöl et al. (2009), who provide a script with their correction algorithm, which is adapted to the correction approach used in the present study. The two-dimensional hill shape was extended perpendicular to the wind direction in order to generate a quasi-two-dimensional flow. Results were taken from the domain centre. The model is run on a grid resolution of either 5 or 10 m, depending on the domain size, which had to be extended for larger hills, to decrease the number of cells.

The third model, Meteodyn WT, is a RANS CFD model, that is able to model roughness and forests (Meteodyn, 2014). It has also a simplified method to account for atmospheric stability. This model is more and more used for wind resource assessments and is considered to be more appropriate for complex and forested sites. More details about the model can either be found in technical reports from the developer (Meteodyn, 2007) or the dissertation of the author (Klaas, 2020). The correction approach described below is implemented and applied to the flow model results from Meteodyn WT. As in WEng, the two-dimensional hill shape was extended perpendicular to the wind direction in order to generate a quasi-two-dimensional flow. A constant horizontal resolution of 10 m was used in a rectangular 'mapping' area in the proximity of the lidar location. Beyond this area, the minimum horizontal resolution drops to 25 m in the outer region of the grid. The vertical resolution decreases with increasing height above ground. However, the first 10 cell layers have a constant resolution of 4 m and then increase by a factor of 1.2 per layer. This results in about 22 m resolution at 150 m and 318 m at the final 35th layer at about 2 km height above ground.

For WEng and Meteodyn WT surface and flow parameters have been varied according to Table 2. Three different roughness lengths were analysed for both models, ranging from smooth bare soil surface characteristics to bushes. For Meteodyn WT the forest model has been used with three heights of 10, 20 and 30 m. Additionally the forest density, i.e. the drag force coefficient $C_d$ has been varied between the three pre-defined settings with $C_d$ equal to 0.001, 0.005 and 0.01 which corresponds to low, medium and high forest density. In any case, the surface and forest parameterization is applied in the whole model domain, i.e. not only on the hill surface but also on its surroundings. Detailed results on this can be found in Klaas (2020).

**Table 2: Roughness lengths $z_0$ of the different roughness maps that were used in the flow models. The purpose of the maps is also given, which includes the corresponding tree heights in Meteodyn WT. Surface characteristics were taken from Troen (1989).**

| Roughness length [m] | Surface characteristics | Tree height [m] | Used in WEng | Used in Meteodyn WT |
|---|---|---|---|---|
| 0.005 | bare soil | - | yes | yes |
| 0.100 | farmland | - | yes | yes |
| 0.500 | bushes, suburbs | 10 | yes | yes (forest and roughness) |
| 1.000 | city, forest | 20 | no | yes (forest) |
| 1.500 | city, forest | 30 | no | yes (forest) |

As shown in Table 3, the atmospheric stability model in Meteodyn WT has also been used to change the stability class which is attributed to a certain Obhukov length $L_*$ in the model. In order to save computation time, this was only done for three selected cases: a low roughness case with $z_0 = 0.005m$, a case with high roughness of $z_0 = 0.5m$ and a forested case with a tree height of 30m and high forest density. This allows to assess the influence of atmospheric stability for different surface characteristics. All parameterizations were modelled for the different hill geometries given above in order to analyse the overall influence for the available combinations of $H/L$ and $z/L$.

**Table 3: Selected atmospheric stability classes from Meteodyn WT (Meteodyn, 2014).**

| Stability class | Stability | $L_*[m]$ |
|---|---|---|
| 0 | Very unstable | -80 |
| 2 | Neutral | 10.000 |
| 6 | Stable | 300 |
| 9 | Strongly Stable | 60 |

## 2.3. Lidar error correction

A definition of the lidar error and its parts can be derived based on the measurement geometry of common Doppler wind lidars. Figure 2 defines the measurement geometry and the coordinate system, as well as the azimuth angle $\theta$ and the half-cone opening angle $\varphi$. It also illustrates the three dimensional measurement geometry with four measurement points. This measurement geometry has been simplified to two-dimensional flow from west to east.

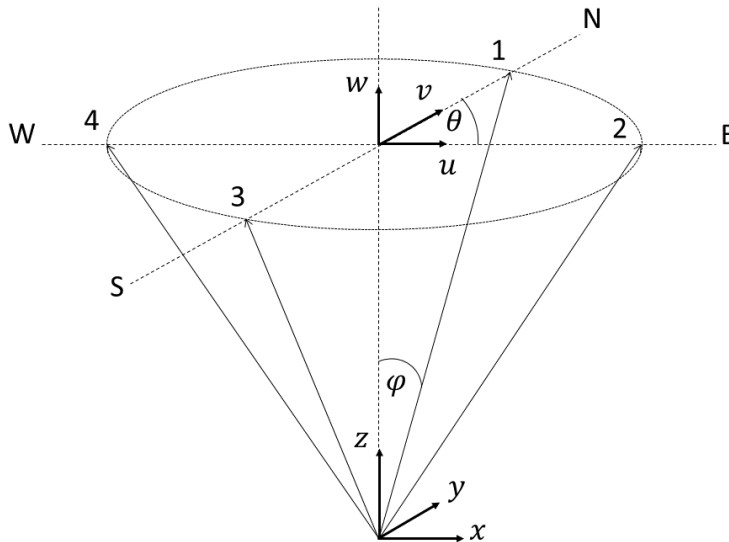

**Figure 2: Lidar measurement geometry and definitions of the local coordinate system $(x, y, z)$ and the wind vector components $(u, v, w)$ as well as the half-cone opening angle $\varphi$ and the azimuth angle $\theta$ (here for the second measurement location, 90° from the north). The measurement locations are numbered starting from North (N) clock-wise to West (W). The measurement locations are shown for an example measurement height at a plane defined by the circle. This measurement geometry equals the one that is used in the Leosphere Windcube v1. In the successive version Windcube v2, a fifths measurement location has been added with $\varphi = 0$ directly above the origin at measurement height.**

The two-dimensional simplification for a lidar placed in flat terrain is shown in Figure 3. The horizontal wind speed is reconstructed with regards to the half-cone opening angle $\varphi$ from the two opposed radial wind speed measurements $v_{r,in}$ and $v_{r,out}$:

$$\hat{u} = \frac{v_{r,in} - v_{r,out}}{2 \sin \varphi} \tag{2}$$

Because the wind speed out the inflow, the outflow and the centre point are equivalent, this results in an unbiased reconstruction of the horizontal wind speed at measurement height above the lidar location.

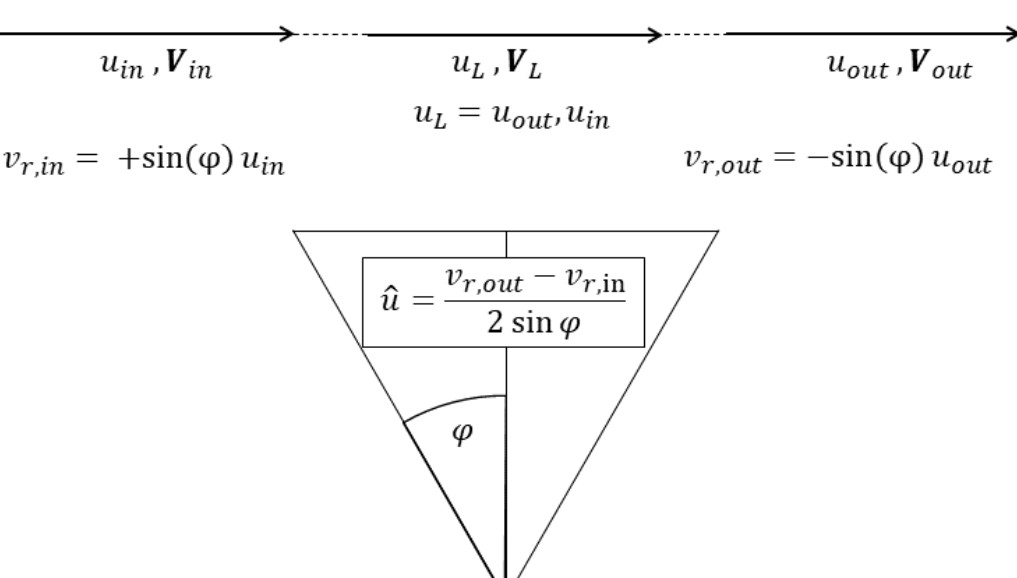

**Figure 3: Generic flow case with a lidar placed in flat terrain in a homogeneous wind field.**

295

Contrary to that, a typical complex terrain flow case is shown in Figure 4 with a lidar placed on top of an arbitrary hill. The reconstructed horizontal wind speed $\hat{u}$ can again be calculated based on the two radial wind speeds and the half-cone opening angle. However, for a given measurement height $z$ it can be noted, that the inflow wind speed at the western measurement point is tilted upwards and the outflow wind speed at the eastern measurement point is tilted downwards. The change in flow

inclination, i.e. in the vertical wind speed component, contributes to the radial wind speeds and therefore introduces an error component. Additionally, due to the speed-up of the wind speed between the two measurement points and the measurement location at measurement height directly above the lidar an additional error occurs.

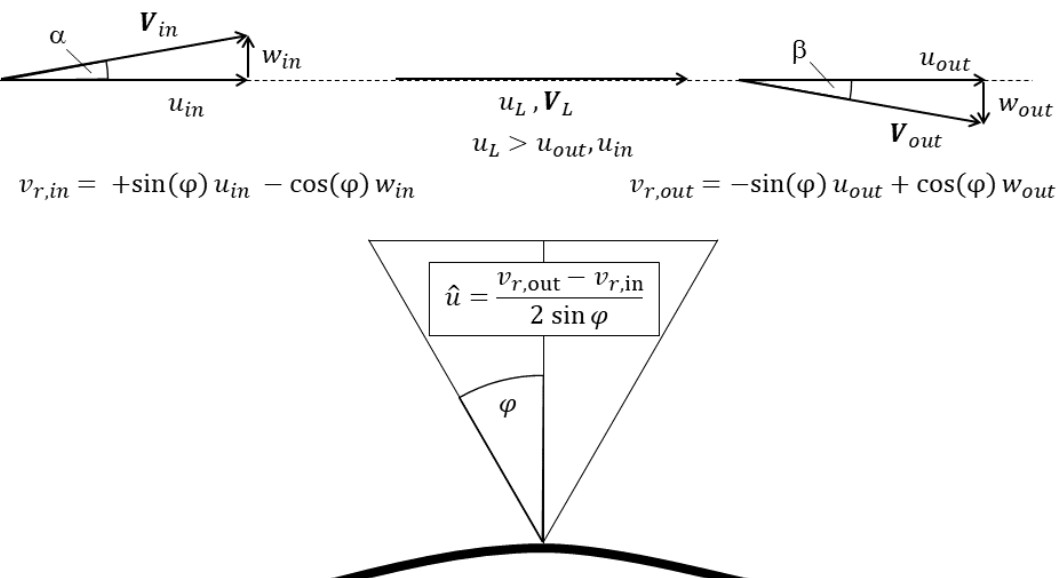

$$v_{r,in} = +\sin(\varphi)\,u_{in} - \cos(\varphi)\,w_{in}$$

$$v_{r,out} = -\sin(\varphi)\,u_{out} + \cos(\varphi)\,w_{out}$$

$$\hat{u} = \frac{v_{r,\text{out}} - v_{r,\text{in}}}{2\sin\varphi}$$

**Figure 4: Generic flow case with a lidar placed on a hill-top. As a typical complex terrain example, this case shows a lidar placed on top of a hill within symmetric flow conditions. The changing vertical wind speed component introduces a lidar error. Additionally, there is a speed-up effect on the horizontal component, which causes a part of the total lidar error.**

Based on the above-given explanations, the lidar error $\varepsilon$ for the two-dimensional case can generally be defined as follows:

$$\varepsilon = \frac{\hat{u} - u_L}{u_L} = \varepsilon_c + \varepsilon_s \tag{3}$$

with $\hat{u}$ being the reconstructed horizontal wind speed and $u_L$ the actual horizontal wind speed at the reconstruction point. Following this definition, an underestimation of the actual wind speed at the reconstruction point will lead to a negative lidar error and an overestimation will lead to a positive lidar error. In order to separate the two effects, the lidar error can be divided into a part being caused by flow curvature ($\varepsilon_c$) and another part due to speed-up effects ($\varepsilon_s$). As presented in the results section, this distinction will give insight into the influence of the half-cone angle on the lidar error.

The equation for wind vector reconstruction can be rewritten for the two-dimensional case as follows (compare Figure 4):

$$\hat{u} = \frac{v_{r,in} - v_{r,out}}{2\sin\varphi} = \frac{V_{in}\sin(\varphi - \alpha) + V_{out}\sin(\varphi + \beta)}{2\sin\varphi} \tag{4}$$

Here the radial wind speeds left and right from the lidar (inflow and outflow) are referenced to as $v_{r,in}$ and $v_{r,out}$ and the magnitude of the wind vector at the same points as $V_{in}$ and $V_{out}$. The inflow and outflow inclination angles of the flow are defined as $\alpha$ and $\beta$ and combined with the half-cone opening angle of the lidar $\varphi$.

For simplification of the above-given equation (4), the following relationship can be derived:

$$u_L = \frac{u_{in} + u_{out}}{2} = \frac{V_{in} \cos\alpha + V_{out} \cos\beta}{2} \tag{5}$$

By making use of this equation and by defining the factor $k = \frac{V_{out}}{V_{in}}$ equation (4) can be written as:

$$\hat{u} = u_L \left(1 - \frac{1}{\tan\varphi}\frac{\sin\alpha - k\sin\beta}{\cos\alpha + k\cos\beta}\right) \tag{6}$$

Neglecting changes in the magnitude of wind speed between the two measurement points here (they will be considered in the second part of the error equation later) by assuming $k = 1$, results in an equation that is independent of the actual wind speed, but only dependent on geometric properties of the wind flow and the lidar:

$$\varepsilon_c \cong -\frac{\tan\dfrac{\alpha - \beta}{2}}{\tan\varphi} \tag{7}$$

And, with $-\alpha = \beta$, as it is the case in symmetrical flow situations the equation reduces to

$$\varepsilon_c \cong -\frac{\tan\alpha}{\tan\varphi} \tag{8}$$

The speed-up of the horizontal wind speed component between a measurement location $i$ and the reconstruction point can be written as

$$\Delta u = u_L - u_i \tag{9}$$

Keeping in mind that the speed-up between both, the inflow and the outflow measurement point and the reconstruction point have to be considered, the lidar error due to speed-up can be defined by

$$\varepsilon_s = \frac{u_{in} + u_{out}}{2u_L} - 1 \tag{10}$$

Here the difference between inflow and outflow horizontal component of the wind flow is considered as $u_{in}$ and $u_{out}$.
In case of symmetric flow with $u_{in} = u_{out}$ the speed-up part $\varepsilon_s$ simplifies to

$$\varepsilon_s = \frac{u_{in}}{u_L} - 1 = \frac{u_{out}}{u_L} - 1 \tag{11}$$

Combining equations (7) and (10) leads to the equation used for the assessment of the total lidar error due to complex terrain in this study:

$$\varepsilon = \left[ -\frac{\tan\dfrac{\alpha - \beta}{2}}{\tan\varphi} \right] + \left[ \frac{u_{in} + u_{out}}{2u_L} - 1 \right] \tag{12}$$

## 3 Results

In the following, the main results of the non-dimensional parameter study are presented, starting with those from the inviscid potential flow model and low surface roughness results for WEng and Meteodyn WT.

The influence of the half-cone opening angle is analysed based on results from the potential flow model and one example from Meteodyn WT. Then the influence of terrain roughness is shown for both, WEng and Meteodyn WT. Finally, exemplary results concerning the influence of forest height and atmospheric stability are discussed. More detailed and complete results can be found in the dissertation of the author (Klaas, 2020).

The lidar error $\varepsilon$ and also its parts due $\varepsilon_c$ (flow curvature) and $\varepsilon_s$ (speed-up) are mostly plotted against the ratio of measurement height over the hill half-width $z/L$. By this, it is possible to extract results for different measurement height as well as different hill dimensions from a single non-dimensional figure. The amount of terrain inclination is in most figures shown for groups of a constant ratio of hill height over the hill half-width $H/L$. In reference to Table 1, mainly four of these groups are analysed from slight slopes up to high slopes in the order of 0.3.

### 3.1.    Influence of orographic complexity and measurement height

Figure 5 (left) shows the results for the lidar error $\varepsilon$ from the potential flow model versus the ratio $z/L$ for four different $H/L$ ranging from 0.1 to 0.4. With increasing $z/L$, the curves follow a distinct shape: The lidar error constantly increases until it reaches a maximum in the range of $z/L$ between 0.5 and 0.6. The exact position is slightly dependent on the $H/L$ ratio and increases with increasing $H/L$. Then the lidar errors starts to decrease for all $H/L$ ratios. Also, the maximum lidar error significantly increases with increasing terrain inclination. For a $H/L$ ratio of 0.1, it is slightly larger than -3 %. For a $H/L$ ratio of 0.4, it reaches up to about -11 %.

Results from the two more sophisticated models WEng and Meteodyn WT for a low roughness length $z_0$ of 0.005 m are given in Figure 6. The black lines indicate the results from the inviscid potential flow model as a reference. The shape and the magnitude of the resulting lidar errors are comparable to those from the potential flow model. However, there are clear differences for both of the models.

The results from WEng (Figure 6, left) show larger lidar errors for $H/L$ ratios of 0.3 and 0.4, i.e. the most complex cases, particularly in the region of $z/L$ between 0.5 and 1. Maximum lidar errors reach up to -12 % for an $H/L$ ratio of 0.4. Results from Meteodyn WT on the other hand show smaller lidar errors for all $H/L$ ratios when compared to the potential flow model. The difference between the two model results increases with increasing $H/L$, and for $H/L = 0.4$ the maximum lidar is about -9.5 %.

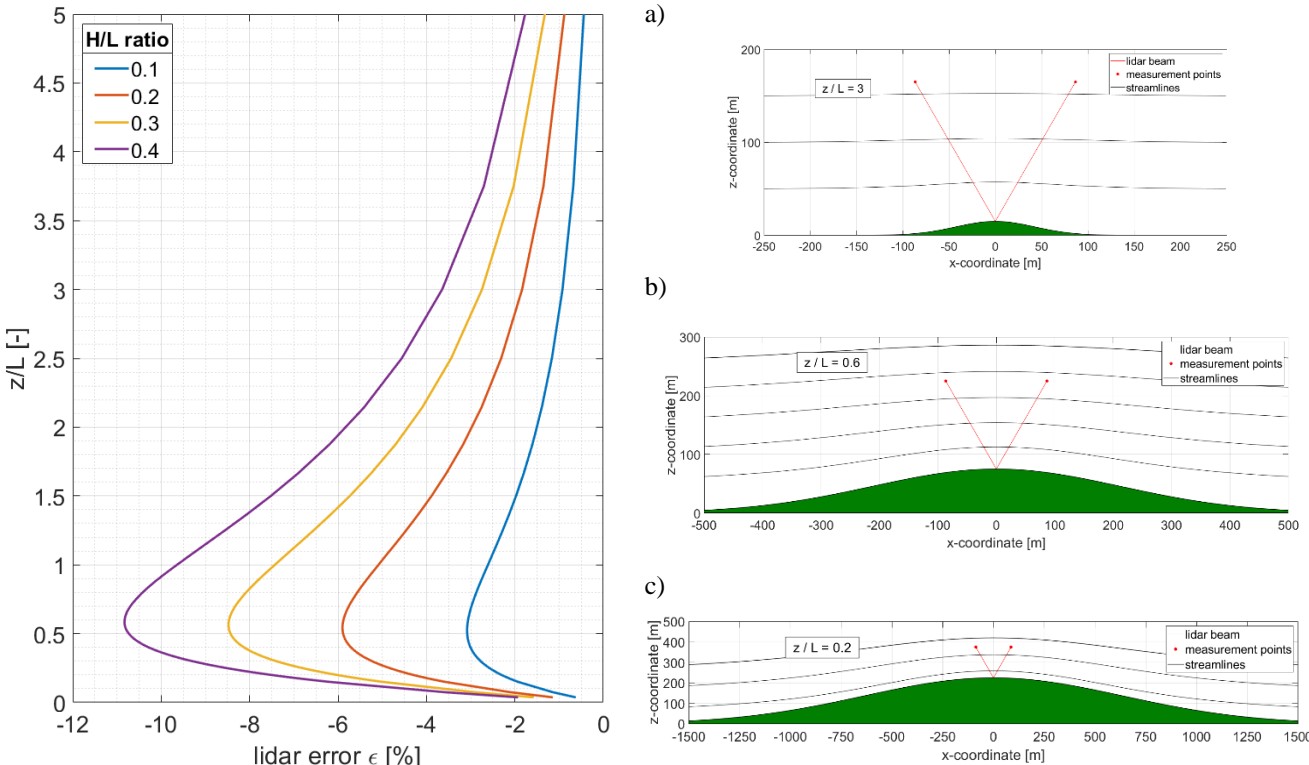

**Figure 5: Left: Lidar error ε in dependence of the ratio $H/L$ between $z/L$ ratios from 0 to 5. Results are based on the potential flow model. Right: Results from the potential flow model for L=50 m (a), 250 m (b) and 750 m (c) for an $H/L$ ratio of 0.3. The lidar position is marked at the top of the hills and the beams are tilted by a half-cone opening angle φ=30°. The measurement points are located at z=150 m above the lidar. The points, therefore, are equal to $z/L$ ratios of 3 (a), 0.6 (b) and 0.2 (c).**

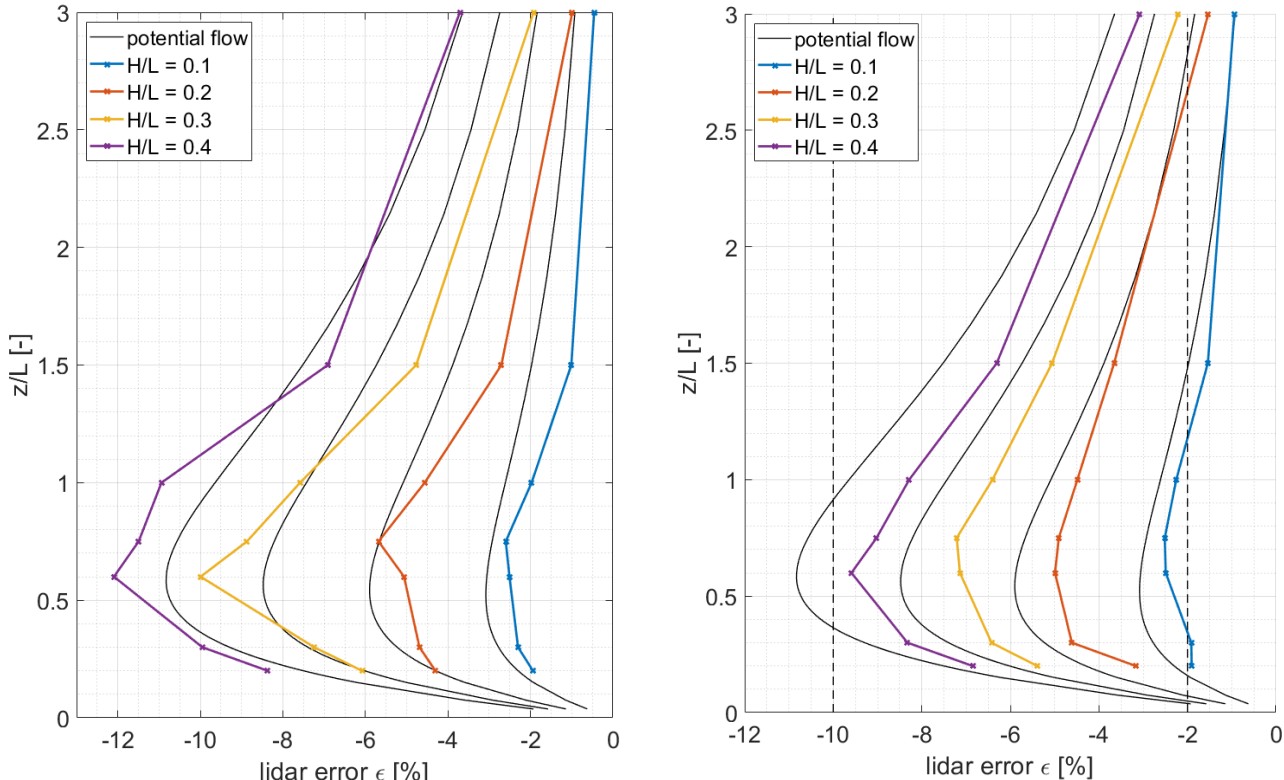

**Figure 6: Lidar error ε in dependence of the ratio $H/L$ between $z/L$ ratios from 0 to 3 for a roughness length of $z_0$ of 0.005 m. Left: Based on results from WAsP Engineering. Right: Based on results from Meteodyn WT. The black lines show results from the potential flow model.**

### 3.2.  Separation of the two lidar error parts

Figure 7 illustrates results from the potential flow model for $\varepsilon_c$ and $\varepsilon_s$ next to each other in the same way that $\varepsilon$ is presented above. The general shape of the four curves is similar to that presented in Figure 5. The maximum errors $\varepsilon_c$ (Figure 7, left) are slightly smaller compared to $\varepsilon$. For an $H/L$ ratio of 0.4. For a $H/L$ of 0.1, the maximum error is about 2.5 %. The maximum error is now located between $z/L$ of 0.45 and 0.51.

Looking at Figure 7, right, which shows the speed-up part $\varepsilon_s$, it becomes obvious that this part is much smaller in magnitude than the curvature part. However, it reaches up to -1.95 % for a $H/L$ ratio of 0.4. The $z/L$ position of the maximum error is between 0.9 and 1.0 and shifts the resulting curves for the total lidar error $\varepsilon$ slightly upwards.

For small $z/L$ the share of $\varepsilon_s$ of the total lidar error is about 10 %. With increasing $z/L$ the share increases up to about 30 %. For large $z/L$, i.e. for relatively small or narrow hills, the speed-up part becomes more important in the overall lidar error correction.

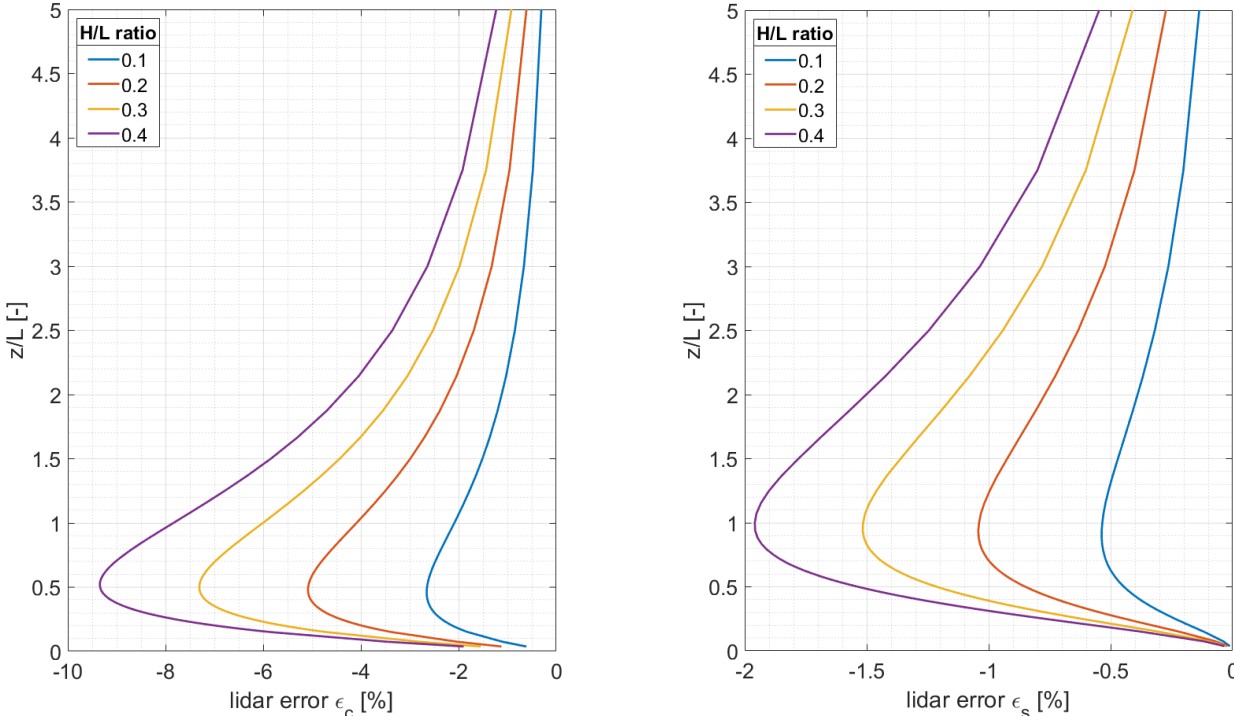

**Figure 7: Lidar error $\varepsilon_c$ (left) and $\varepsilon_s$ (right) in dependence of the ratio $H/L$ between $z/L$ ratios from 0 to 5. Results are based on the potential flow model. Note the different scaling of the x-axes.**

### 3.3.    Influence of the half-cone opening angle

Figure 8 is illustrating the influence of changing the half-cone opening angle $\varphi$ of the lidar measurement geometry on $\varepsilon$, $\varepsilon_c$ and $\varepsilon_s$. Results are presented for the potential flow model first, in order to analyse and explain the basic principle.

Figure 8 (left) shows the lidar error $\varepsilon$ for the four $H/L$ ratios and the three different half-cone opening angles 30°, 20° and 10°. When decreasing the half-cone angle, there is also a slight decrease in the lidar error. This is particularly true for $z/L$ ratios between 0.3 and 1.0, which is in the range of the maximum lidar error. For larger $z/L$ ratios, the influence of $\varphi$ on the lidar error decreases. For small $z/L$, the influence is also only marginal.

Figure 8 (middle and right), which shows the lidar error split up into $\varepsilon_c$ and $\varepsilon_s$, enables to retrace of the individual contribution of flow curvature and speed-up effects on the total lidar error $\varepsilon$. It becomes obvious that decreasing the half-cone opening angle significantly decreases $\varepsilon_s$. While $\varepsilon_s$ reaches up to -2 % for an $H/L$ ratio of 0.4 for a $\varphi$ of 30°, it falls below -0.25 % for all $H/L$ ratios for an angle of 10°.

Decreasing $\varphi$ has a contrary influence on $\varepsilon_c$. While there is almost no influence for the lowest two $z/L$ ratios, there is an increase of $\varepsilon_c$ for half-cone angles of 20° and 10° when compared to the original 30°. This difference is largest for $z/L$ above 0.5 and persists up to $z/L$ of 3.

The superposition of the two opposing effects results in the total effect on $\varepsilon$ as it is shown in Figure 8 (left).

Figure 9 provides an example result based on Meteodyn WT with three different half-cone opening angles for an $H/L$ ratio of 0.3 and a roughness length $z_0$ of 0.005 m. Here results for both, the 30° and 20° angle are approximately the same. However, a further decrease to 10° results in smaller lidar errors for $z/L$ ratios below 0.6 and larger lidar error above.

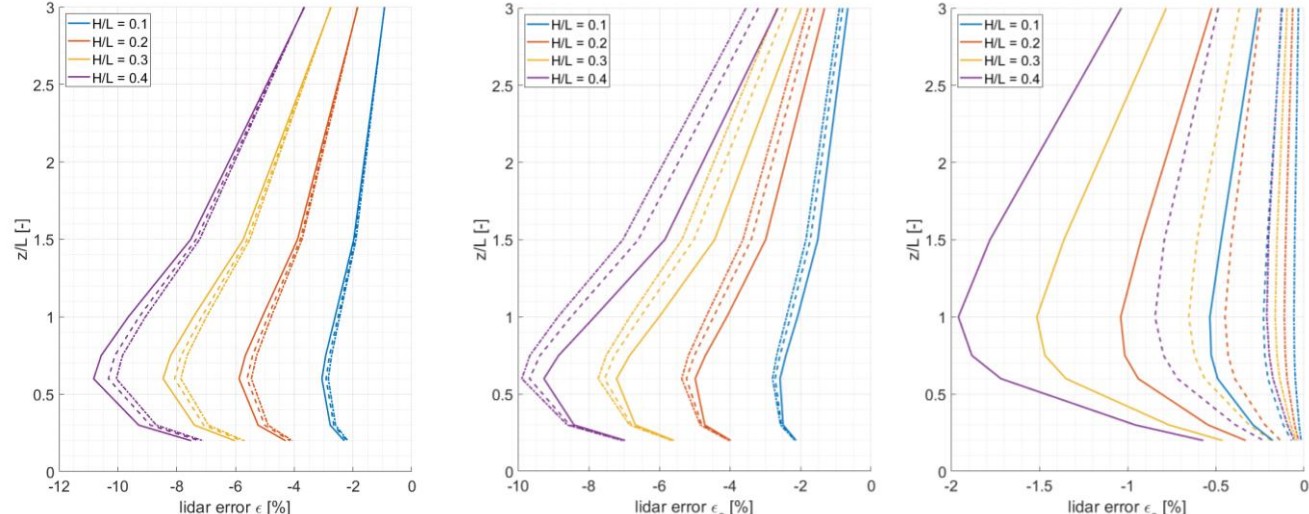

Figure 8: Lidar error $\varepsilon$ (left), $\varepsilon_c$ (middle) and $\varepsilon_s$ (right) in dependence of the $H/L$ ratio for the half-cone opening angles $\varphi$ of 30° (solid lines), 20° (dashed lines) and 10° (dot-dashed lines) between $z/L$ ratios from 0 to 3 (left). Results are based on the potential flow model.

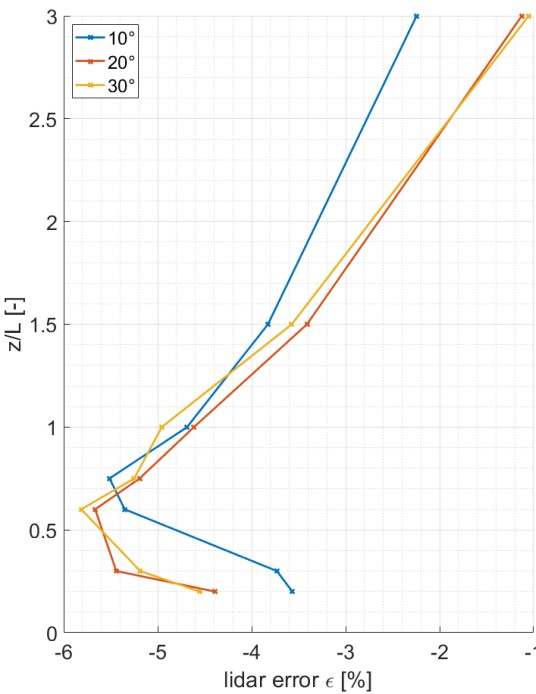

**Figure 9: Lidar error $\varepsilon$ for an $H/L$ ratio of 0.3 for the three different half-cone angles $\varphi$ of 30°, 20° and 10° between $z/L$ ratios from 0 to 3 (left). Results are based on Meteodyn WT for a roughness length $z_0$ of 0.5 m.**

**3.4.    Influence of surface roughness**

The influence of roughness length on the lidar error has been analysed for the results from WEng and Meteodyn WT for the three different used values of 0.005 m, 0.1 m and 0.5 m. They are exemplarily shown for an $H/L$ ratio of 0.3 in Figure 10. The general shape of the lidar error curve is comparable to that from the potential flow model. However, the maximum lidar error based on the WEng simulations exceeds that found with the potential flow model. Here the maximum value found at $z/L$

of 0.6 is about -10 %, which is significantly larger than in the potential flow model, especially for the lowest roughness. The estimated lidar error decreases strongly for smaller and larger $z/L$.

Figure 10 (right) shows the influence of the roughness length $z_0$ in Meteodyn WT. While the curve for the lowest roughness length has a pronounced maximum value and the shape of the curve is comparable to that from the potential flow, this is no

longer the case for higher roughness length. For a $z_0$ of 0.1 m, the lidar errors are generally smaller for all $z/L$ ratios, except the lowest two 0.2 and 0.3. For these, the model results are very close to each other. At the maximum point, the lidar error is decreased by about one percentage point for the medium roughness length by another percentage point for all $z/L$ ratios. However, the general shape of the error curve stays similar.

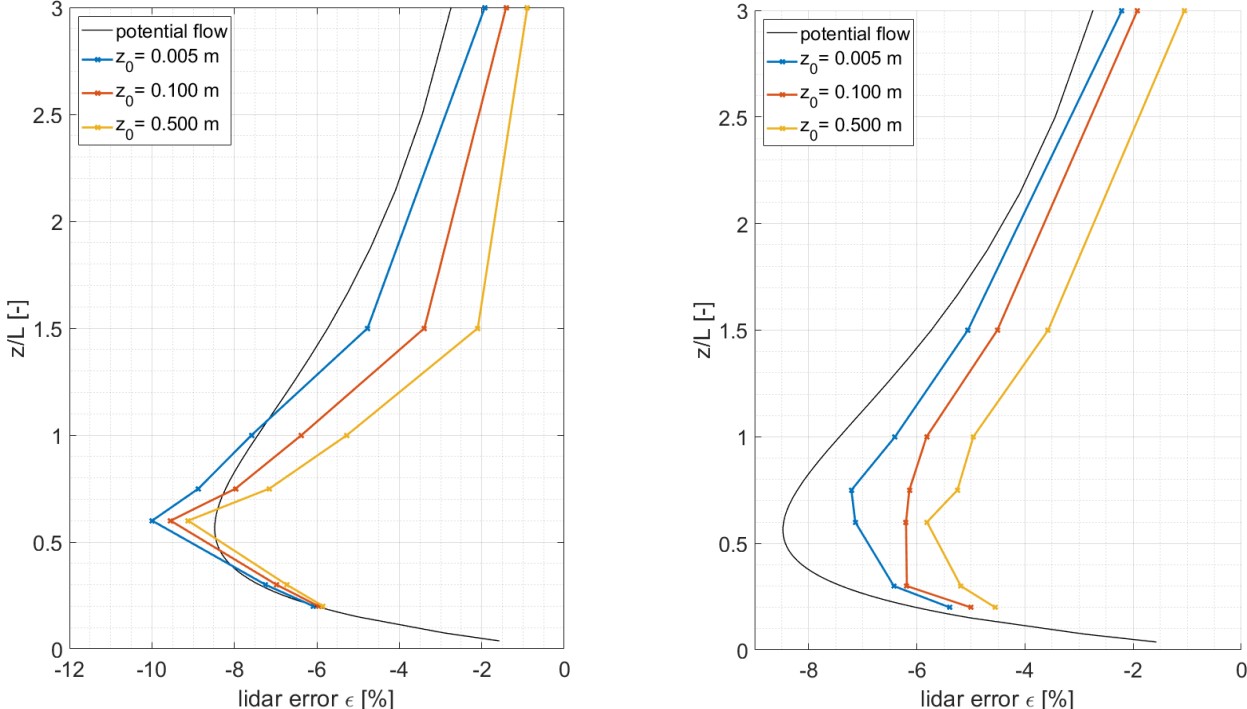

**Figure 10: Model comparison between WAsP Engineering (left) and Meteodyn WT (right) for different roughness lengths for a $H/L$ ratio of 0.3. Lidar error $\varepsilon$ in dependence of the roughness length $z_0$ between $z/L$ ratios from 0 to 3. Results from the potential flow model (black) are shown as a reference.**

### 3.5. Influence of forest

Figure 11 shows the impact of different tree heights on the total lidar error. The results are shown for a $H/L$ ratio of 0.3, however the influence of the forest on the lidar error is different, depending on terrain inclination.

The total lidar error ε shows a significant dependence on tree height. Highest lidar errors are found for small tree heights of 10 m and. When increasing the tree height to 20 m and 30 m, ε decreases for all $z/L$ ratios but the largest. Although maximum values for the lidar error can still be seen around $z/L$ ratios of 0.6, the shape of the curves is not entirely comparable with that

from the potential flow model as a reference.

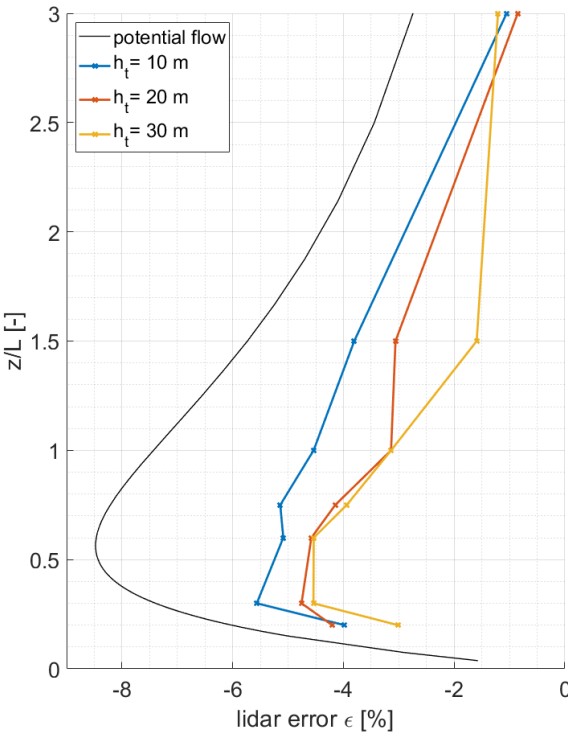

**Figure 11: Lidar error ε in dependence of tree heights $h_t$ between $z/L$ ratios from 0 to 3. Results are based on Meteodyn WT (colored) for medium forest density and the potential flow model (black) for an $H/L$ ratio of 0.3. .**

### 3.6.    Influence of atmospheric stability

In order to analyse the influence of different atmospheric stability conditions, the stability class in Meteodyn WT has been modified for a part of the simulation cases. The influence of the stability parameter is most severe for medium to large $H/L$ ratios. The results presented in Figure 12 sum-up these results for the four chosen stability classes very unstable (0), neutral (2), stable (6) and strongly stable (9) (compare Table 3), exemplarily for a forested case and an $H/L$ ratio of 0.3. These four out of ten possible stability classes in Meteodyn WT cover the whole possible range. Calculations have been carried out for all stability classes in-between those four, but the effects found are systematic and it is, therefore, sufficient to show only this excerpt.

A clear tendency of reduced lidar errors for increasing atmospheric stability can be seen. Largest lidar errors occur for very unstable stability conditions, with a clear maximum at a $z/L$ ratio of 0.6 and a lidar error ε of about -6 %. Coming to neutral, stable and strongly stable cases, the maximum is again shifted towards lower $z/L$ ratios. The maximum error for strongly stable cases is -2.3 % and can be found at a $z/L$ ratio of 0.3. Below and above the maximum, there is a relatively sharp decrease

in lidar errors, which pronounces the maximum points. For the largest $z/L$ ratio of 3.0, the influence of atmospheric stability is relatively small.

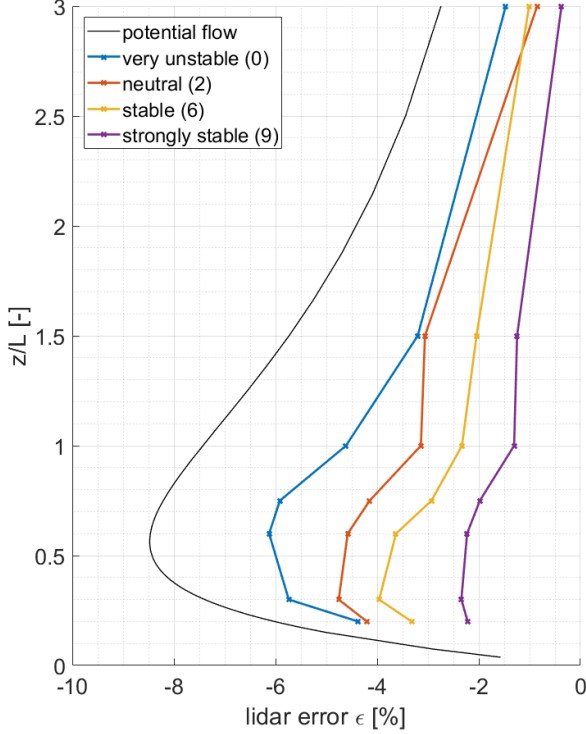

Figure 12: Lidar error ε in dependence of atmospheric stability between $z/L$ ratios from 0 to 3. Results are based on Meteodyn WT (colored) and the potential flow model (black) for an $H/L$ ratio of 0.3. Results are shown for the forested case with a tree height $h_t = 20\ m$ and medium forest density.

## 4 Discussion

As described in section 2.2, this study is focussed on the complex terrain error of a lidar placed at the top of a Gaussian hill. This simple setup allows to illustrate and discuss the influence of terrain complexity and model parameterization in a systematic way. A key finding is that regardless of the model and its parameterization as well as the complexity of the hill all results show that lidars underestimate the wind speed at measurement height above the lidar location. In other words, a convex flow pattern in the proximity of the lidar causes an underestimation of the wind speed. Contrary to that, concave flow (e.g. in valleys) will result in an overestimation of the wind speed at the measurement location.

Obviously, lidars are not always placed on top of a hill in real-world applications. Moving the lidar location to the flanks of the hill will influence the resulting lidar error because both, the inflow and outflow angles at the measurement points and the speed-up will change. For flow fields symmetric to the hill centre, moving the lidar upwind or downwind will change the lidar error, which can be demonstrated e.g. with the potential flow model (compare Bradley (2008)). However, for more complex flow situations the streamlines are no longer symmetric. E.g. for forested cases their turning points shifts downwind (Figure 13 left), causing a more complex interconnection between the lidar position and the resulting errors.

Beyond that, real complex terrain has an arbitrary, three-dimensional structure. Effects from surrounding hills, valleys, escarpments as well as changes in surface properties will influence the flow field at the lidar measurement location. In addition, the flow field changes with changing wind direction in the three dimensional case. Because of this flow complexity, it is necessary to run full three-dimensional wind flow models for all relevant wind directions to get a useful lidar error estimation. The overall terrain effects can e.g. be analysed in a "lidar error map" which shows the lidar errors for a given measurement height above the terrain. An example for this is given in Klaas et al. (2015).

However, such complex approach are not suitable for fundamental parameter studies as presented here. However, based on the simple, two-dimensional Gaussian hill study it is possible to analyse the influence of main parameters of the measurement setup as well as the type and parameterization of the flow models used to estimate the lidar errors.

As already described in section 3.1, the right-hand side of Figure 5 illustrates three selected Gaussian hills with an $H/L$ ratio of 0.3 and a measurement height $z$ of 150 m. Based on this, the relationship between measurement height and hill size, i.e. hill half-width can be analysed in detail. Increasing $z/L$ ratios can be interpreted in two ways:

First, the hill size decreases for increasing $z/L$ ratios when keeping the measurement height constant. For a given measurement height the maximum lidar error is then found for $L$ in the range of 1.6 to 2.0 times $z$. Lidars errors decrease strongly for broader hills (smaller $z/L$) and also for more narrow hills (larger $z/L$). For a typical measurement height of 150 m, the corresponding hill half-width is about 250 m for a hill height of 75 m at the point of maximum lidar error.

Second, when keeping $L$ constant, increasing $z/L$ reflect an increasing measurement height. From this perspective, lidar errors are relatively small for low measurement heights and increase strongly up to a $z$ equal to 50-60 % of $L$. Beyond the maximum point, lidar errors decrease continuously until they become insignificant as terrain influence on the wind flow diminishes. For the above given hill ($L = 250, H = 75m$), the lidar error is still about -2 % for a hypothetical measurement height of 600 m and reaches its maximum for a measurement height in the order of 150 m which is around hub height of modern wind turbines.

Deduced from these observations, the shape of the lidar error curves can be explained as follows: For very low measurement heights (or very broad hills) both, the local speed-up effects and the flow inclination angles are small and so are the resulting lidar errors. With increasing $z/L$, both effects significantly increase, because either the measurement point distance increases with increasing measurement height or – from the perspective of hill size – the hill becomes more narrow and flow inclination and local speed-up effects in the vicinity of the lidar increase. Then, with $z/L$ increasing above 0.6, the terrain influence

diminishes for increasing measurement heights. From the perspective of terrain scaling, for narrow hills, the lidar measurement points move left and right towards the flanks of the hill and therefore to positions where flow inclination decreases (compare Figure 5 a). For very small hills, additionally speed-up effects become less important. The lidar error separation shown in

Figure 7 also illustrates the importance of both, flow curvature and speed-up effects in dependence of $z/L$.

From a practical point of view, the above findings and considerations urge to carry out lidar error estimation for a given site for all relevant measurement heights. Depending on the actual site characteristics, the lidar error might well be dependent on measurement height. However, it is also possible that a height dependency is not visible in the data as for example in Foussekis (2009). The reason for this could be that uppermost measurement height is only 100 m where the actual $z/L$ ratios are within

a range of only small changes in lidar error with height or that the height dependency is superimposed by other flow features.

These above discussed findings are supported by all three applied flow models, which show comparable behaviour for increasing $z/L$ ratios (Figure 6). Both of the more sophisticated models (WEng and Meteodyn WT) show a comparable shape of the error curves in dependence of $z/L$. However, while WEng tends to give larger maximum errors for $H/L$ ratios of 0.3

and 0.4 than the potential flow model, Meteodyn WT shows smaller errors for all hill geometries. In comparison to the potential flow model the other two models are run with a small roughness length $z_0$ of 0.005 m. Besides the general characteristics of the different flow models, this difference in parameterization might be a reason for the deviations between the models. Additionally, there is a general tendency of WEng to over-predict speed-up effects in complex terrain, which is well known in the literature (Bingöl et al., 2009; Foussekis, 2009). The reason for this is, that the tendency for flow separation in the lee of

the hill is not predicted by WEng as it assumes attached wind flow (Bowen and Mortensen, 1996). This tendency becomes more important for increased terrain complexity, i.e. increased $H/L$ ratios. However, there is no such tendency for the RANS CFD model Meteodyn WT and the consideration of surface roughness influences the flow patterns above the hills, which leads to decreased lidar errors.

It is important to put the magnitude of the estimated lidar errors into the context of the overall uncertainties of wind measurements (regardless of the measurement technology used) and the total uncertainties of wind resource assessments. As already discussed in the introduction, e.g. the German Technical Guideline 6 on Wind Resource Assessment states, that an additional uncertainty for the correction of lidar errors in complex terrain should be considered that is 50 % of the estimated errors (FGW e.V., 2017). In practice, this approach provides an upper limit for terrain complexity where lidar measurements

are still reasonable. However, it is difficult to provide exact values for maximum tolerable lidar errors since other uncertainties in the wind resource assessment have to be considered as well. On the other hand, also mast-based cup and sonic anemometers, which are often used for wind energy applications, are prone to increased uncertainties at complex terrain sites. For example increased turbulence and flow inclination increase cup anemometer uncertainties (Dahlberg et al., 2006).

Surface roughness is found to have a significant effect on the wind flow patterns over a hill and therefore changes the magnitude of the lidar error estimations. In both models – WEng and Meteodyn WT – increasing the roughness length results in decreasing lidar errors (Figure 10). Additionally, the presence of forest significantly decreases the estimated lidar errors. Both effects can be explained by the increasing asymmetry of the hill flow (Belcher et al., 2012; Ross and Vosper, 2005). In particular, the critical slope for flow separation is reduced by the influence of the forest (Ross and Vosper, 2005). Streamlines for three

different tree heights are shown in Figure 13 (left). It becomes obvious that the flow pattern on the lee side of the hill changes dramatically with the streamlines being shifted upwards. The effect is most severe for tall trees. But, in comparison to the potential flow model results, already small trees of 10 m height have a significant effect on the wind flow. Increasing the roughness length has comparable effects, although they are not as severe as effects from the forest. Generally, it can be deduced that both, increasing surface roughness and the presence of forest decreases the lidar errors. This also explains that the FCR

method from Leosphere, described in section 1, overestimates the lidar error in complex terrain, in particular for high roughness or forested cases (Wagner and Bejdic, 2014; Leosphere, 2017).

Additionally, it should be pointed out that the lidar error estimation is very sensitive to roughness and forest parameterization, i.e. roughness length, tree height and forest density. Detailed knowledge of the actual surface characteristics is needed to fit the model results to the measurement site. This becomes obvious e.g. Klaas et al. (2015), Figure 6, showing a range of results

for different forest parameterizations and the resulting lidar errors at a complex terrain site. In comparison to the measured data, it becomes clear that it is mandatory to take into account these effects in lidar error estimation. Neglecting forest (or roughness) effects will result in severe overestimation of the actual lidar error. This is counterintuitive, because increased terrain complexity is usually considered to increase measurement uncertainties.

Atmospheric stability influences flow patterns above or around hilly terrain (Leo et al., 2016). The present model based study provides results on the influence of stable and unstable flow in comparison to neutral flow conditions (Figure 13, right). Stable stratification suppresses the vertical exchange and therefore hinders the streamlines from being shifted upwards in the lee of the hill (Ross et al., 2004; Emeis, 2018: 86). Speed-up in stable conditions as well as flow inclination angles in the vicinity of the lidar increase which results in increased lidar errors. On the other hand, unstable stratification has contrary implications on

the wind flow patterns. On the lee side of the hill the streamlines are elevated which is most severe for the "very unstable" case shown in Figure 13. This results in decreased speed-ups and smaller (or even positive) flow inclination angles on the downwind side of the hill. In this flow pattern the lidar errors are reduced. However, the influence of atmospheric stability on the actual lidar error in complex terrain needs more investigation and the result presented in this study can only be seen as a starting point for this. Different atmospheric conditions are often related to specific wind directions. It is therefore very difficult

to separate the effects of stability from those caused by complex orography, roughness or vegetation.

Generally, real complex terrain sites are characterized by a combination of different influencing parameters. As an example, atmospheric stability will have different effects on the actual lidar error at a forested site than at a site without forest. As forest

increases the tendency for flow separation, this will most likely be more important in unstable situations – or could be
suppressed in stratified flow conditions (compare Figure 13, right). For an accurate estimation of the lidar error at a given
complex terrain site it is therefore necessary to gather complete information on local weather and atmospheric condition
statistics as well as reliable surface data. This will help to provide a detailed terrain model and to setup model parametrization
in a way that covers all relevant wind conditions. The results presented in this study will help to prioritize available model
parameters and to decide on whether it is necessary to consider e.g. atmospheric stability or to acquire more accurate forest
data.

The separation of the total lidar error $\varepsilon$ into the two parts $\varepsilon_c$, induced by flow curvature and $\varepsilon_s$, induced by local speed-up
effects is presented in section 3.2. To the knowledge of the authors there is no comparable study in the literature so far. The
study therefore allows for an in-depth analysis and discussion of the contribution of these two effects for the first time.
Splitting up the lidar error shows that flow curvature is responsible for the main part, causing 70 to 90 % of the total error,
slightly dependent on the $z/L$ ratio. Figure 7 exemplarily shows the error separation for results from the potential flow model.
In Klaas (2020) the error parts for the two other models and different parameterizations are analysed as well. Results are
comparable and indicate that the flow curvature induced error always is the main part of the total error. However, based on the
overall results it becomes clear that lidar error estimation methods should consider both effects: Flow curvature and speed-up
to achieve accurate prediction of the lidar errors. Both error parts have the same sign, resulting in an overall larger total error
when the speed-up part is considered as well.
Splitting up the total lidar error, allows to discuss and understand the effects of smaller half-cone opening angles of the lidar
geometry than the default 30°. However, results for the total lidar error $\varepsilon$ for three different half-cone opening angles (10, 20
and 30°) show only small differences in the potential flow model (Figure 8). Looking at the error parts $\varepsilon_c$ and $\varepsilon_s$ reveals the
complex dependency between the half-cone opening angle and the resulting lidar errors. While the speed-up part diminishes,
because the measurement points move closer together, the flow curvature part is increased by approximately the same amount.
The measurement points are moved to a location where flow inclination angles are slightly larger, which causes $\varepsilon_c$ to increase.
In total, the two parts cancel out, resulting in almost no influence of the half-cone opening angle on the lidar error in symmetric
flow conditions. Contrary to that, results from a forested (and therefore asymmetric) case in Meteodyn WT in Figure 9 shows
that a reduction of the half-cone opening angle can have significant influence on the total lidar error. Here, a reduction from
30 to 10° reduces the lidar error at $z/L = 0.3$ by about 1 percentage point and causes the same increase at $z/L = 3$.
The present study relaxes the assumption of a linear change in vertical wind speed that is used in Bingöl et al. (2009) and
allows for more complex flow pattern with nonlinear and asymmetric changes in the vertical wind speed, as it can e.g. be
observed in forested and stable flow cases (compare Figure 13). This also results in lidar error equations that include the half-
cone opening angle as a relevant factor (see section 2.3). Based on these findings it might be interesting to study the possibility
of optimizing the half-cone opening angle for specific sites and measurement heights in order to further reduce lidar errors in
complex terrain. In any case, the results emphasize the need for correction methods that include the half-cone opening angle.

Simplified methods or models, based on the assumption of symmetric flow are insufficient to cover the effects of forest or stability at complex terrain sites.


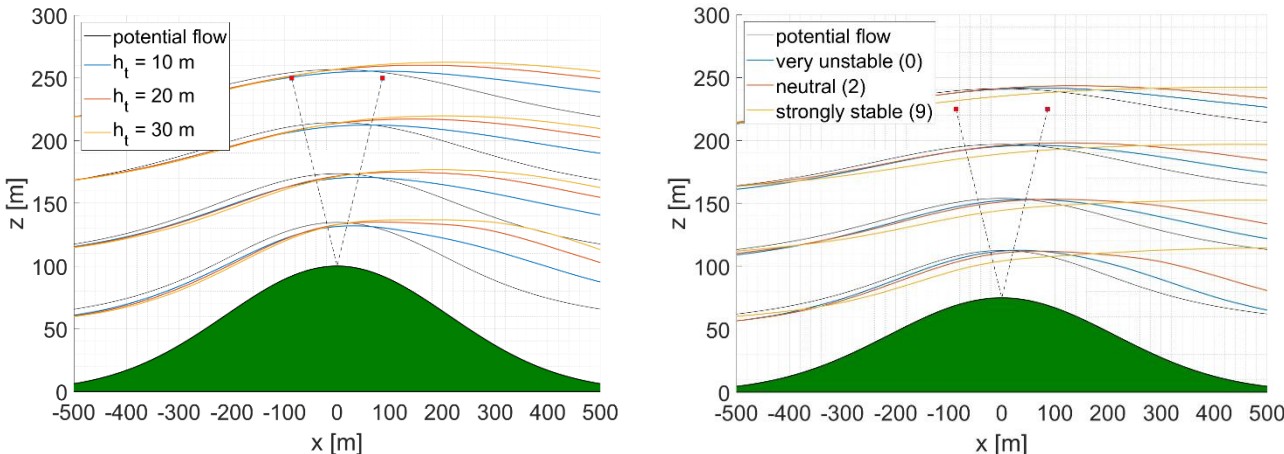

**Figure 13: Streamlines in dependence of tree height $h_t$ (left) and atmospheric stability class (right) for an $H/L$ ratio of 0.3. Streamlines from Meteodyn WT (coloured) and from the potential flow model (black) are starting at $z$ of 50, 100, 150 and 200 m in front of the hill. Lidar measurement points at 150 m measurement height with a $\varphi$ of 30 ° are marked in red.**

A complete validation of all results of this study would be a challenging task. A possible approach for validation would be to find an isolated hill or ridge with a comparable shape and erect a tall reference mast on top of it. However, diverse model parameterizations for roughness and forest would make it necessary to validate the results on many different sites that reflect the actual parameters. Beyond that, variable atmospheric stability conditions would be essential to validate the wind flow modelling and the lidar error estimation for these situations.

These requirements for a validation project are hard to meet. A general validation of the applicability of the used flow models can therefore be a first step to support the findings with regards to lidar error estimation. A validation of the potential flow model is carried at by Bradley (2012). For two complex terrain sites the results show a comparable quality of error estimation as with more detailed RANS CFD models. However, as there is no consideration of surface roughness, vegetation, flow separation or atmospheric stability influence on the wind flow, the applicability of the potential flow model is limited.

WEng and WAsP are extensively used for wind energy application for more than 20 years (Mann et al., 2002). Validations show that the accuracy of wind flow modelling with WAsP is strongly dependent on the orographic complexity of the terrain. Generally speaking WAsP should only be applied in neutral atmospheric conditions and gentle terrain where no flow separation occurs (Bowen and Mortensen, 1996). Additionally, there is no forest model in WEng. Although the estimation of speed-ups and wind shear can be improved by the introduction of a displacement height, more complex effects, such as enhanced flow

separation caused by the forest cannot be resolved (Dellwik et al., 2006). Validations for Meteodyn WT show that e.g. the speed-up effects for high slopes and rough surfaces can be estimated with more accuracy by such a model (Ayotte, 2008). The

flow model has been tested against measurement data e.g. from Askervein hill showing a good agreement (Meteodyn, 2007). However, a detailed further discussion of the flow models is beyond the scope of this study. An extensive review of flow over complex terrain is for example given in Finnigan et al. (2020), also including an overview over numerical modelling.


## 5 Conclusion and outlook

Following a non-dimensional approach, the study allows for a comprehensive analysis of the effects of orographic complexity, measurement height, terrain roughness, forest, atmospheric stability as well as the half-cone opening angle of the lidar device on the resulting lidar error $\varepsilon$. Results and conclusions shown here do not represent the complete contents; the reader is therefore

encouraged to study the full text of the dissertation for more and detailed information. Nonetheless, the presented results allow for a broad overview on the estimation of lidar errors in complex terrain by the use of flow models.

Unsurprisingly, the orographic complexity of the terrain, i.e. the $H/L$ ratio of the considered hill geometries has by far the most influence on the resulting lidar error estimations. For all models and parameterizations, the lidar error significantly

increases with increasing $H/L$ ratio. Depending on the flow model used, the lidar error becomes about 4-5 times larger when increasing the $H/L$ ratio from 0.1 to 0.4.

Beyond that, the study introduces a novel approach that separates the total lidar error $\varepsilon$ into its parts $\varepsilon_c$ and $\varepsilon_s$ which are caused by flow curvature and local speed-up effects. Based on this concept, it becomes clear that under most circumstances both parts

contribute significantly to the total lidar error. In any case, the major part is caused by flow curvature, i.e. $\varepsilon_c$. However, the actual share of $\varepsilon_s$ is dependent on the $z/L$ ratio and 10-30% of the total error can be attributed to local speed-up effects between the probe volumes. Resulting from this, error correction approaches for Doppler lidar profilers should always take into account both effects to minimize uncertainty in error estimations. Simplified approaches that rely solely on flow curvature or inclination might very well underestimate the total lidar error for specific terrain properties.


The non-dimensional concept of this study allows for a comprehensive assessment of the influence of measurement height on the lidar error. Initially, lidar errors strongly increase with increasing measurement height because the distance between the measurement points increases. Flow curvature and local speed-up effects come into play and both contribute to the total error. On the other hand, for very large measurement heights, the influence of the terrain on the overall flow field diminishes. As a

trade-off between these two effects, maximum lidar errors occur at a $z/L$ ratio of approximately 0.6. This complex interaction between measurement height and the actual terrain shape makes a detailed preliminary assessment of lidar errors for a measurement campaign at a given site and for the planned measurement heights mandatory.

Flow model parameterization has shown to be crucial for the estimation of lidar errors. This applies for terrain roughness, forest height and density as well as atmospheric stability and consequently for any combination of these. Generally speaking, rough and forested terrain decreases lidar errors. Correction approaches should therefore include forest models. From the results it can be concluded that, the potential flow and linearized flow models should only be applied for unforested sites with small low terrain inclinations. Under the presence of forest and especially for steep terrain inclinations, those models will significantly overestimate lidar errors, because they are not able to capture effects from e.g. flow separation in the lee of the hill. Atmospheric stability, in particular stable stratification, has significant influence on the lidar error estimation. Given a site with strong variations in atmospheric stability, this should also be considered in the lidar error estimation approach. However, these findings need to be researched in more detail in the future, especially under consideration of measurement data and information about atmospheric stratification.

In literature, the half-cone opening angle is usually considered to have negligible effects on the lidar error (Bingöl, 2009; Foussekis, 2009; Bradley, 2012). However, the concept of separating the total lidar error into a flow curvature and a speed-up part, it is possible to assess the impact of changing the half-cone opening angle in more detail. The analysis reveals opposed effects of decreased half-cone opening angles on $\varepsilon_c$ and $\varepsilon_s$. This explains the small influence on the total error that has been observed in other studies.

As an overall summary, it can be concluded, that the findings of this study clearly show that orographic complexity, roughness and forest characteristics, as well as atmospheric stability, have a significant influence on lidar error estimation. This study provides helpful guidance on the choice and parameterization of flow models as well as on the design of methods for lidar error estimation. The results emphasize that the use of a RANS CFD model in conjunction with an appropriate forest model is crucial to achieve reasonable lidar error estimations in complex terrain. If atmospheric stability variation at a measurement site plays a key role, the influence on the flow characteristics will also significantly affect the lidar error at those sites and should be considered in the modelling. In the context of a wind resource assessment, an accurate estimation of the prospective lidar errors should be carried out before the measurement campaign. It is then possible to make an early decision on whether a lidar measurement is feasible at the given site.

However, for a better validation of the findings, a broader basis of measurement data would be beneficial. Ideally, a measurement campaign with the specific purpose to validate the key findings of this study could be designed at and around a forested hill. The measurement site should be carefully chosen under consideration of the dimensions $H$ and $L$ of the hill parallel to the prevailing wind directions. Furthermore, the possible $z/L$ ratios should be examined, so that the dependence of the lidar error on height can be validated.

In particular, with regards to atmospheric stability, three-dimensional terrain could enable an investigation of more complex flow patterns in different stability situations. Especially for stable stratification, where the flow could tend to stream around the hill, rather than over it, the results might be different for three-dimensional terrain (Leo et al., 2016).

Additional and more complex flow features such as flow separation, which might occur in very complex terrain situations, have not been treated within the context of this study. For this, a non-stationary flow model could be used to analyse the influence of periodic recirculation phenomena behind escarpments. Such a model, together with more advanced turbulence modelling, could also help to explain the scatter that occurs in ten-minute values of lidar measurement errors in real-world applications.

## 6 Acknowledgements

This paper is based on the detailed research within the doctoral thesis of the author (Klaas, 2020), which would not have been possible without the valuable discussions, advice and encouragement with Prof. Stefan Emeis who supervised the thesis. As the first author of this paper, I would like to thank him very much for his support, patience and continuous enthusiasm throughout this period. I am also grateful for the support and feedback from my co-worker Dr. Doron Callies when preparing the dissertation and this paper. Additionally we would like to thank Michael Courtney and the second referee for their extensive constructive feedback on this paper.

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
