# Peer review of "The five main influencing factors on lidar errors in complex terrain"

_Wind Energy Science, 2021_

## Referee Comment (RC1)

Review of: Kaas and Emeis, The five main influencing factors on lidar errors in complex terrain.

General

I applaud the perusal of insight and simplicity clearly demonstrated by this paper. Particularly the idea to split the error source into two parts, curvature and speed-up, is well seen and well demonstrated. Similarly I like the structuring with models of increasing complexity and the opportunity this gives to draw insight and identify limitations.

I am a little disappointed however in the low amount of insight that is drawn and lack some deeper reflections on the implications of the findings. My main suggestion is therefore to add a 'Discussion' section (not necessarily with this name and (please) separate from the Conclusions) where the authors reflect on (suggestions):

- That the errors (on hill tops?) are always negative. Is this a general result, why and can we expect lidar measurements in positions where it wouldn't be true?
- The reasons for the trends that are seen (this is already tackled to some degree in the text). For example  (one only that caught my eye), why does increasing stability give larger errors where increasing roughness (and forest) gives smaller errors?
- How do the parameterizations combine? (e.g. if you have strongly stable flow and tall trees, are the predicted errors even lower?)
- (Why) Should we believe these results? To what extent is there experimental corroboration?
- Would other 'hi-fi' models give similar results?
- It's a bit alarming to see how much the errors change with the parameterizations? What does all this say about the uncertainty of the corrections?

The elegance of the paper would be heightened by a corresponding lift in the quality of the writing. Not that this is bad but some of the explanations are more difficult to follow than necessary. Perhaps use more examples (measurement situations) in explaining the trends shown in the different plots. Make the text shorter where you can.

Specific comments:

Abstract
First para:
Wind lidars (or Doppler lidars) don't have much to do with "light detection and ranging" – they don't detect or range!! It's just a name so please don't labour this.  Is it the 'measurement principle' (what is this?) that is the problem – or the assumptions inherent in the processing of the raw data?

L15: suggest remove e.g.

L20: suggest remove *manifold* (I had to look it up and it's my mother tongue..)

Last sentence:

*'When planning a measurement campaign, an accurate estimation of the prospective (predicted?) lidar error should be carried out in advance **to decrease measurement uncertainties** and **maximize the value**.':*

Knowing what the error isn't enough of course – you need to make the correction in order to reduce the uncertainty. What are you maximizing the value of?

1 Introduction

L32: *Due to their principle of measurement,..* – Please be more specific here

L32: profiles = profilers

L45: *equivalent* – change to identical?

L59: *there is no significant increase in the* magnitude of the *errors* (add magnitude of the)

L77: delete between

A general comment here is that most, if not all, of the literature fails to recognize that traditional instrumentation (i.e. cups) in complex terrain have a fairly high uncertainty.

L134: Do you expect (or not) that stability directly affects the lidar accuracy? (I think this is most unlikely). Understand your explanation that stability indirectly affects the error through changed profile, turbulence(?) and flow patterns. Are there other flow parameters that might be relevant- e.g. Froude number?

L139: *Courtney et al.* **were** *(not* **are***) proposing…* They probably got wiser from the subsequent experiments…;)

2 Methods

L184: The uncertainty of all experimental data should be estimated before comparing to models – cups as well as lidars. How should this uncertainty be used?

L191: Again, what about the 'mast' measurement uncertainty?

L222: win -> wind

L249: I think you are using 'exemplary' wrongly throughout the paper. Please check its definition. I would write '*are shown as examples within this study*'.

L268: Are these results only valid for a top-placed lidar? What about the more general case of a lidar on the side of a hill (where the error is usually lower anyway)?

L272: and error -> an error

L283: add *horizontal* after *reconstructed*

L293 (eq 3) – please state here what u_in and u_out are (you do this later in L310). Are there assumptions (symmetry?, linear change in speed?) in this simplification? I wasn't completely clear about this.

L302 (eq 6) – why, contrary to expectations, does the opening angle still appear here?

L315 mostly -> usually

L315: why does this comment appear here and not in the results section?

L343: Several comments regarding the -2% threshold for cup-anemometers and the comparison with this. Firstly is 2% realistic (the reference is in German, but it is easy enough to find typical values for cup operational uncertainties (class B in complex terrain) and add in mounting and calibration uncertainties. I would suspect higher values than 2% (was this using class A?). And is this standard uncertainty (k=1)?

But this would be the total uncertainty for the cup. Wouldn't it be fairer to compare with the total uncertainty of the lidar and not just the uncertainty due to the non-homogeneous flow?

The -10% seems very arbitrarily chosen. Do we expect an uncertainty of 10% if the error is 10%?

L351: All of this section takes some digestion. Try and make the text more straightforward, E.G (L351) *For the least complex hill, ….* I would suggest something like *For the least complex hill (H/L=0.1), the lidar error is below 2% until a non-dimensional height (z/L) of 0.16 is reached. The error reaches a maximum of 3% at z/L=0.5 and falls below 2% again above z/L= 1.5.* Maybe even set this in context, e.g. with a hill L=1000m and H=100m.

3.4 Influence of surface roughness

L427: *reference* – which reference?

L436: The -2% dashed line is missing in Figure 7 (Right)

---

## Referee Comment (RC2)

Review: The five main influencing factors on lidar errors in complex terrain, wes-2021-26

General comments:

The authors are discussing one of the critical issues in lidar measurements, namely the error associated with lidar measurements. In particular, this contribution is focusing on the fundamental assumption of flow homogeneity which is the cornerstone of horizontal wind reconstruction techniques. This work split the error into two parts, which are very promising in understanding and deeply analysing the source of errors.

Having said that, this work lacks in many places, especially the discussion section, providing deep insights about the predicted results. Moreover, the newly introduced error classification has been ignored in many of the result subsections. That has been reflected on the conclusion section as well.

Here is a list of general suggestions and points which could improve the manuscript:

- Improve the description of the model setups in order to improve the reproducibility of the work.
- Provide an insightful explanation to the predicted trends in the error.
- Add where it is possible as a ratio of the total error, the two error parts.
- Add where it is possible more than one model results.
- Although turbulence plays an essential rule in atmospheric flow, it is ignore in this analysis. Examining turbulent intensity and/or turbulent kinetic energy could give some insights about the predicted results.
- The current study depends on steady state flow models which quite understandable for the scope and the applications of this work, but the authors could at least discuss this effect. In the end, Lidars do not measure stationary wind conditions.
- Although the error estimation is relative to the model full predicted wind vector, a validation of the model setups will be beneficial. The hill case is well studied in the literature, and there is a great chance to find data for some of the modelled cases in this work.

The specific comments section is reflecting most of the above comments. In general, the manuscript language could be improved in some places, but nothing major. The citation format is not consistent across the paper, for instance sometime it takes the form of Author (Year) or (Author, year) or (Author (year)). Please follow the journal format and be consistent across the manuscript. Same applies on the figures, for instance -2% dashed line is missed in some plots.

Specific comments:

0.  Abstract:

The abstract could be improved by including some key results of the study, such as the error increase value due to one or two parameters. Also, adding the conclusion about the error due to the cone angle, since it is one of the main finding of the study and explains the contradictions in the literature.

L11: influencing factor on or of

L14: reformulate to make it more clear

L15: remove ,eg.,

1.  Introduction:

L32: Group citation should be in chronological order

L34: I guess you mean the horizontal wind vector/speed reconstruction

L42: chapter -> section

L47: This statement is quite general, the assumption is not valid when there is a significant spatial changes between the measurement points. Therefore within the same complex site, it will depend on the position of the profiler lidar and the cone angle.

L49:L51: it is not clear if you are comparing Anotoniou et al. (2007) with Smith et al. (2006), and what is the main contradiction between the two. Please reformulate.

L52: This 5-10% error reported by Courtney et al. (2008) is common in complex terrain, please elaborate from that study why? Otherwise it is not clear for the reader why reducing the cone angle from 30deg -> 15deg is necessary.

L59: It is great to mention this contradiction but please elaborate more and mention why there was no significant increase in the error.

L67: not dependent -> independent . I agree with this conclusion but with larger cone angle, there is a higher change to capture inhomogeneous flow.

L71: why is it independent of height ?

L86: do you mean RANS CFD here? Please be more specific, CFD is a general term.

L114: Group citation should be in chronological order

L123: Group citation should be in chronological order

L135: Indeed stability has a significant impact on wind profiles and turbulence levels even in flat terrain, but it is not obviously clear to the reader at this point how it could influence the lidar accuracy. Please elaborate more.

L137: is another factor -> do you mean 'are another factors' ?

L145: Group citation should be in chronological order

2. Methods

I suggest to have some subsections here, at least two, one for the model and one for the lidar error.

L178: forest parameters are not necessarily surface parameters, for instance a typical forest model parameters have to be mapped as volume values.
L191: Group citation should be in chronological order
L196: Isolated one sentence should not be a separate paragraph. Please reformulate it.
L200: The literature is full of similar cases for Gaussian and Cosine hills and the similarity parameters are well established. Please give a background about this case and cite the appropriate literature. Also add the mathematical formula of the hill or cite the source.
L200: H and L should be formatted as math for consistency.
L222: win flow → wind flow
L240: The used models are steady, aren't they? Add this to the description of the models
L241: Add the forest parameters, (low, medium and high) are very general terms
L241: is the forest applied uniformly above the ground? Or limited to the hill. Please elaborate more on the forest model setup.
L247: This was done for a low roughness case, ……. It is not clear what do you mean here.
L249: exemplaryly ?
L250: The model description lack any information about the used meshes. For instance, in addition to the mesh horizontal resolution and domain size, the first cell size is an important parameter especially with high roughness values.
L255: One could start a subsection here.
L286: chapter → section
L303: Eq(6), is this assumption applied in this work?
L311: equation 6 → equation 5?
L313: Are equation (9) and (1) equivalent? In other terms, if one estimated the error based on Eq(1) and Eq(9) will get the same value or not? Have the authors tested that?
L313: Is the same approach applicable om three-dimensional flow? If so, it would be beneficial to end this part with the general formulation.

3. Results and discussion
L323: exemplaryly →exemplary?

Subsection 3.1:
Fig 4: for the right hand side figures a, b, c, it is more reasonable (and more common for such case) to use normalized x, and y axis with respect to L or H. Also the choice of y-coordinate is confusing since the vertical component is considered z-dir in the rest or the paper.
L378: speep-up → speed-up
L378: what about the error variation due to the model? Each model has a different error prediction and it would be good to show it for this case.
Subsection 3.2:
L380: Add to the text the used model, it is not enough to mention it in the figure caption only.

L380: I think this subsection is essential for this study because it shows the two error parts. So it is essential to show the effect of the model on the error distribution. For example, the speed-up error ratio (with respect to the total) should be relatively higher in RANS model compared to potential flow model. Showing the ratio of each part across the three models will enrich the discussion.

L395: is the predicted flow of all cases and models fully attached? Although it could happened downstream directly above the ground, it may influence the measurement heights.

Subsection 3.3

I have similar comments as above, why it is limited to one model here. Especially, potential flow is symmetric and does not show the expected diverse behavior in the curvature and speed-up errors.

Subsection 3.4

L425: as the above mentioned comments, without including the other models in comparison to the potential flow, it would be hard to isolate the influence of the roughness. The increase in the max error here is not because of the roughness only.

L436: This subsection lacks any explanation for why the error decrease which increasing the roughness values. Probably it is related to turbulence mixing.

L436: it would be interesting to see the influence of roughness on the ratio of the two errors.

Subsection 3.5

As mentioned in my comments on the method section, the forest parameters are not clear. Did you use the same LAD or LAI?

L445:446: epsilon is not math format

Fig8: I do not see the point of including the forest and stability in the same figure.

Fig8: there is no explanation for this behavior, why the error decrease with increasing the forest height, or why both 20m and 30m cases have the same max error.

L451: again this section focuses on the total error and does not show the two error parts which are the main promise of this work.

Subsection 3.6

It is not mentioned what are the model parameters behind these stability classes. For instance, Table 3 includes three stable conditions, which on is included here?

Same comments as before, what is the effect of thermal stratification on the two errors? Any insights why does the error decrease with moving towards stable condition?

4. Conclusion

It is highly advised to revisit this section once applying the above comments. It could benefit also from adding more insights about this future work, it could be a subsection or one paragraph.

---

## Author Comment (AC1)

**Review of: Klaas and Emeis, The five main influencing factors on lidar errors in complex terrain.**

The authors would like to thank the referee very much for the valuable and concise feedback provided and the effort he put into this. We appreciate the comments and suggestions very much. They are very helpful to increase the quality of the manuscript and add relevant information for the reader.

In the following, we answered all of the referee's questions, comments and suggestions. We state how we use the feedback in the revised version of the manuscript and provide additional information and explaination for the referee whereever needed.

**General**

I applaud the perusal of insight and simplicity clearly demonstrated by this paper. Particularly the idea to split the error source into two parts, curvature and speed-up, is well seen and well demonstrated. Similarly I like the structuring with models of increasing complexity and the opportunity this gives to draw insight and identify limitations.

I am a little disappointed however in the low amount of insight that is drawn and lack some deeper reflections on the implications of the findings. My main suggestion is therefore to add a 'Discussion' section (not necessarily with this name and (please) separate from the Conclusions) where the authors reflect on (suggestions):

- That the errors (on hill tops?) are always negative. Is this a general result, why and can we expect lidar measurements in positions where it wouldn't be true?

- The reasons for the trends that are seen (this is already tackled to some degree in the text). For example (one only that caught my eye), why does increasing stability give larger errors where increasing roughness (and forest) gives smaller errors?

- How do the parameterizations combine? (e.g. if you have strongly stable flow and tall trees, are the predicted errors even lower?)

- (Why) Should we believe these results? To what extent is there experimental corroboration?

- Would other 'hi-fi' models give similar results?

- It's a bit alarming to see how much the errors change with the parameterizations? What does all this say about the uncertainty of the corrections?

The elegance of the paper would be heightened by a corresponding lift in the quality of the writing. Not that this is bad but some of the explanations are more difficult to follow than necessary. Perhaps use more examples (measurement situations) in explaining the trends shown in the different plots.

Make the text shorter where you can.

We would like to thank you very much for your detailed and well structured general feedback on our manuscript. We are pleased to read your general appreciation on the topic and how we tackled it. Beyond that, we find your suggestions on a 'Discussion' section very helpful. After having studied all of your feedback and suggestions and reviewed our submitted manuscript, we agree with you that it would highly benefit from a separate 'Discussion' section. We have therefore added that section between the 'Methods' and 'Conclusions' sections. In the revised manuscript we have moved already

existing discussion paragraphs to the new section. We have extended our discussion according to the following points, which also include your above given feedback:

- Discussion on general findings for all parameters, which includes explainations on when the lidar error is negative (or positive) and about the implications of measurement locations on a hilltop, on the flanks of the hill etc.
- Discussion and interpretation of the error parts and their importance
- Discussion of the role of the half-cone opening angle, including references to relevant literature on that topic
- Discussion and interpretation of the influence of roughness, forest and atmospheric stability including explaination of the reason for the influence whereever possible
- Discussion of the implications of the sensitivity of the lidar error estimation to model parameterization and what this means for applications
- Discussion of combinations of different parameters (roughness and stability or forest and stability etc.)
- Considerations regarding the magnitude of the lidar error in comparison to e.g. cup anemometry in complex terrain
- Considerations on the flow models used, including possible limitations and discussion about other 'hi-fi' models
- Discussion on available (and needed) experimental validation of the flow modelling results

We have included additional literature references where needed to support our discussion. We have also included additonal plots with streamlines from the RANS flow model that help discussing and interpreting the influence of roughness (forest) and atmospheric stabilty on the model results.

**Specific comments and answers:**

**Abstract**

First para:
Wind lidars (or Doppler lidars) don't have much to do with "light detection and ranging" – they don't detect or range!! It's just a name so please don't labour this. Is it the 'measurement principle' (what is this?) that is the problem – or the assumptions inherent in the processing of the raw data?

We agree that in the relevant literature it is common to simply use the term „lidar" or „wind lidar". We have therefore changed the first sentence and have removed the explaination of the abbreviation.

The second sentence was meant to briefly state that wind lidars are prone to errors at complex terrain sites, which is then elaborated throughout the manuscript. To be more precise here, we propose to write: „However, because of the assumption of homogeneous flow in their wind vector reconstruction algorithms, common Doppler lidars suffer from errors at complex terrain sites."

We have rephrased this in the revised version of the manuscript.

L15: suggest remove e.g.

We removed „e.g." in the revised manuscript.

L20: suggest remove *manifold* (I had to look it up and it's my mother tongue.

We have replaced „manifold" by the more common term „various" which we think fits better here.

Last sentence:
*'When planning a measurement campaign, an accurate estimation of the prospective (predicted?) lidar error should be carried out in advance* **to decrease measurement uncertainties** *and* **maximize the value**.*'*:
Knowing what the error isn't enough of course – you need to make the correction in order to reduce the uncertainty. What are you maximizing the value of?

Thank you for your comment.

Of course, you are right, that besides the estimation of the lidar error it is neceassary to apply a correction to the actual measurement data. However, the idea of error estimation **before** the measurement campaign is to choose reasonable measurement locations where lidar errors are relatively low. E.g. in the German FGW Technical Guideline 6 – „Determination of wind potential and energy yields", the additional uncertainty due to lidar error correction is assumed to increase proportionally to the magnitude of the lidar error.
Because of this, it is beneficial to choose a measurement location with low predicted lidar errors that will then only add small additional uncertainties to the overall wind ressource assessment.

We agree that „predicted" fits better to the context than „prospective" and have changed this in the revised version of the manuscript.

We rephrased the last two sentences to clarify why lidar error estimation should be carried out before the measurement campaign in order to optimize the choice of the measurement location and that the overall aim is to maximize the value of the measurement data by decreasing measurement uncertainties.

**1 Introduction**

L32: *Due to their principle of measurement,..* – Please be more specific here

Similar to the Abstract, we have rephrased this sentence to make it more specific.

L32: profiles = profilers

Thank you, we have corrected this in the revised manuscript.

L45: *equivalent* – change to identical?

We agree that „identical" is the more appropriate term and have changed it in the revised manuscript.

L59: *there is no significant increase in the* magnitude of the *errors* (add magnitude of the)

Thank you, we have corrected this in the revised manuscript.

L77: delete between

Thank you, we have corrected this in the revised manuscript.

A general comment here is that most, if not all, of the literature fails to recognize that traditional instrumentation (i.e. cups) in complex terrain have a fairly high uncertainty.

Thank you for your comment.

Indeed we agree that flow in complex terrain also has an influence on cup anemometer (or sonic anemometer) uncertainty. E.g., increased turbulence and flow inclination differ from those in flat terrain, which leads to higher uncertainties. A procedure to account for the acutal ambient conditions is therefore implemented into the IEC 610400-12-1 standard, Appendix I. From relevant studies we learned that class indexes significantly increase for class B sites (e.g. J.-Å. Dahlberg, Friis Pedersen, T., and P. Busche: ACCUWIND - Methods for classification of cup anemometers, Denmark. Forskningscenter Risoe. Risoe-R, 2006).

In the revised manuscript we have added a short paragraph to the introduction that discusses calibration and classification uncertainties of cup anemometers in complex terrain. We have also added relevant literature about this. Additionally we are discussing the magnitude of the lidar errors from our parameter study in the context of measurement uncertainties in the newly introduced 'Discussion' section.

L134: Do you expect (or not) that stability directly affects the lidar accuracy? (I think this is most unlikely). Understand your explanation that stability indirectly affects the error through changed profile, turbulence(?) and flow patterns. Are there other flow parameters that might be relevant- e.g. Froude number?

Thank you for your comment.

We expect atmospheric stability to change the flow patterns above complex terrain. Changed flow patterns will then affect the lidar errors, which makes the lidar error indirectly dependent on atmospheric stability. We support this with relevant literature on flow over complex terrain in neutral and stratified atmospheric conditions (e.g. Ross, A. N., Arnold, S., Vosper, S. B., Mobbs, S. D., Dixon, N., and Robins, A. G.: A comparison of wind-tunnel experiments and numerical simulations of neutral and stratified flow over a hill, Boundary-Layer Meteorol, 113, 427–459, https://doi.org/10.1007/s10546-004-0490-z, 2004.).

We have added this to the newly introduced 'Discussion' section. To explain better, how flow patterns change due to changes in atmospheric stability, we have added a plot with streamlines over a hill from our simulations.

The Froude number might be used to classify atmospheric stability. In our study we use the Obukhov length as a measure for atmospheric stability. This measure is applied by the CFD model Meteodyn WT that we use in our study. It is also possible to calculate the Obukhov length from high frequency sonic anemometer measurements, which often makes it the method of choice to classify atmospheric stability. We have not done any investigations on other flow parameters such as Froude Number or Bulk Richardson number and their influence on flow pattern and therefore lidar error. A more detailed discussion of stabiliy classification methods is beyond the scope of our study as we think and might be part of future work.

L139: *Courtney et al.* **were** *(not* **are***) proposing…* They probably got wiser from the subsequent experiments…;)

We are certain about that and have rephrased the sentence in the revised manuscript.

**2 Methods**

L184: The uncertainty of all experimental data should be estimated before comparing to models – cups as well as lidars. How should this uncertainty be used?

Thank you for your comment.

The intention of this paragraph is to point the reader to the fact that there are uncertainties in all relevant aspects: The flow model predictions, the lidar measurement data and also the specific aspect of lidar error correction. So either – when no lidar data correction is applied – the uncertainty of the data is considered to be higher at a complex terrain site due to (unknown) lidar errors. Alternatively, after application of a lidar error estimation and correction, the uncertainty can be reduced. Still an additional uncertainty from the lidar error correction has to be considered.

We have slightly rephrased the paragraph in the revised manuscript to clarify our intention.

The uncertainty should be used to evaluate the comparison between e.g. modelled and measured wind profiles at the complex terrain site. If lidar errors significantly contribute to the overall measurement uncertainty, it might not be possible to validate model results without the application of the error correction.

L191: Again, what about the 'mast' measurement uncertainty?

Yes, we agree that also the mast has an (increased) uncertainty at a complex terrain site. We added this into the above mentioned paragraph in the revised manuscript and the newly introduced 'Discussion' section. See also our comment above.

L222: win -> wind

Thank you, we have corrected this in the revised manuscript.

L249: I think you are using 'exemplary' wrongly throughout the paper. Please check its definition. I would write '*are shown as examples within this study*'.

Thank you very much for your advice. We have checked and corrected or rephrased this throughout the revised manuscript.

L268: Are these results only valid for a top-placed lidar? What about the more general case of a lidar on the side of a hill (where the error is usually lower anyway)?

Thank you for your comment.

All results presented in the study are valid for a lidar placed on the top of the hill. We have added a sentence in this paragraph to clarify this. We agree that this is a special case and that real measurement locations will often be on the flanks of a hill, on a plateaus, close to escarpments or in valleys.

However, considering all of these cases would expand the study very much. It is also not trivial to answer the question if errors on the flanks of the hill are generally lower than on the top. E.g. when considering a forested case, the streamlines are shifted downwind. The turning point of the flow (where positive vertical wind changes to negative vertical wind) will be found in the lee of the hill. Having a lidar placed on the downwind site of the hill will then eventually lead to larger errors than on the top.

A valuable tool to analyse lidar errors for different is the lidar error map that is presented in Figure 7 of Klaas et al. (2015) https://doi.org/10.1127/metz/2015/0637 . We have added a reference to this in the newly introduced 'Discussion' section and now provide a more elaborate discussion on other cases in the revised manuscript.

L272: and error -> an error

Thank you, we have corrected this in the revised manuscript.

L283: add *horizontal* after *reconstructed*

Thank you, we have added „horizontal" in the revised manuscript.

L293 (eq 3) – please state here what u_in and u_out are (you do this later in L310). Are there assumptions (symmetry?, linear change in speed?) in this simplification? I wasn't completely clear about this.

Thank you for you comment.

$u_{in}$ and $u_{out}$ refer to $u_4$ and $u_2$ in Figure 3, which have their indices from the clockwise numbering in the three-dimensional case (Figure 2). We agree that it is confusing to use different notations in the figure and the text / equations. We have therefore updated Figure 3 in the revised manuscript and now only use subscripts 'in' and 'out' as in the following equations and text.

There are two main assumption that are used when deriving the equations:

1. The ratio $k$ between outflow and inflow wind speed ($k = V_{out}/V_{in}$) is set to 1, assuming that both wind speeds are the same. This removes $V_{out}$ and $V_{in}$ from the equation.
2. The horizontal wind speed at the lidar location $u\_L$ is defined as the mean of $u_{in}$ and $u_{out}$. This makes it possible to cancel $u_L$ out of the equation, when inserting equation 4 into the lidar error definition equation 1 (left part).

These two assumptions enable us to derive an equation for the flow curvature induced lidar error $\varepsilon_c$ that is only dependent on inflow angle $\alpha$, outflow angle $\beta$ and half-cone opening angle $\varphi$.

With respect to these assumptions, we have slightly changed the resulting equations (5) and (6) and replaced the equality sign by an approximately equal sign.

(!) Please note that we have introduced an additional equation in the revised manuscript – the Gaussian hill definition – that is now equation (1). Following equation numbers have therefore been shifted by 1. Above given equation numbers refer to the original manuscript.

L302 (eq 6) – why, contrary to expectations, does the opening angle still appear here?

As described in the manuscript the error equations are based on a very general approach and only two minor assumptions (see above) are used.

Contrary to other simplified approaches of lidar error correction (especially Bingöl, F., Mann, J., and Foussekis, D.: Conically scanning lidar error in complex terrain, Meteorologische Zeitschrift, 18, 189–195, 2009.), the cone-opening angle does not cancel out in our approach. This is because we do not assume the vertical wind speed to change linearly, which is the main assumption in Bingöl's considerations.

We have added this to the newly introduced 'Discussion' section to emphasize the difference to already available literature.

L315 mostly -> usually

We agree and have changed this in the revised manuscript.

L315: why does this comment appear here and not in the results section?

We agree that this should be moved to the results section as it describes the way the results are presented. We have changed this in the revised manuscript.

L343: Several comments regarding the -2% threshold for cup-anemometers and the comparison with this. Firstly is 2% realistic (the reference is in German, but it is easy enough to find typical values for cup operational uncertainties (class B in complex terrain) and add in mounting and calibration uncertainties. I would suspect higher values than 2% (was this using class A?). And is this standard uncertainty (k=1)?

Thank you very much for your comment. We agree that measurement uncertainty with cup anemometry at complex terrain sites might have higher total uncertainties than 2% (standard uncertainty, k=1). The intention of drawing the -2% and the -10% lines was to raise awareness about several aspects:

1. Cup anemometers as a reference or standard sensor, which is mostly applied in wind energy, do also have uncertainties (of course) and we wanted to roughly indicate their magnitude.
2. With complex terrain induced lidar errors in an order of 2 %, the total uncertainty of a wind ressource assessment will not benefit much from a correction.
3. The overall uncertainty of wind ressource assessments should stay below approximately 15% for complex terrain sites. Lidar errors in the order of 10% will add too much to the uncertainty, even if a correction is applied (see next comment). Therefore, lidars are not a good choice at such complex sites.

However, based on your comments we have decided to remove the -2% and -10% lines from the plots, because we think fixing these values will not be helpful for the discussion of the results in the manuscript. Instead, we have added a more elaborate discussion about mast / cup anemometer uncertainties at complex terrain sites (class B) in the newly introduced 'Discussion' section. We have also added relevant (and English) literature for this discussion We hope that this will help the reader to better interpret the magnitude of the complex terrain lidar errors.

But this would be the total uncertainty for the cup. Wouldn't it be fairer to compare with the total uncertainty of the lidar and not just the uncertainty due to the non-homogeneous flow?
The -10% seems very arbitrarily chosen. Do we expect an uncertainty of 10% if the error is 10%?

Thank you very much for your comment. Several aspects of this comment are already answered above. We have removed the -10% line in the revised version of the manuscript.

Nevertheless, we would like to explain why we have chosen -10%: From literature (which we have now added to the revised version of the manuscript) we found that total uncertainties of about 15% at complex terrain sites can be seen as an upper limit for a reasonable (bankable) wind ressource assessment. Without consideration of the additional uncertainty due to lidar error correction, we fixed the total uncertainty of a wind ressource assessment at a complex terrain site to 12 %, which is based on literature research and rough estimations of the specific uncertainties within a wind ressource assessment in complex terrain. Additional uncertainties caused by lidar errors will now increase the total uncertainty of the wind ressource assessment. Following the relevant FGW Guideline 10, we expect an additional uncertainty of 50% of the lidar error (5% uncertainty at 10% error). For a magnitude of 10% of the lidar error the total uncertainty will increase beyond 15%.

Therefore, we think that lidar measurements are no longer feasible at such sites. We have added a more elaborate discussion on this in the revised version of the manuscript.

L351: All of this section takes some digestion. Try and make the text more straightforward, E.G (L351) *For the least complex hill, ….* I would suggest something like *For the least complex hill (H/L=0.1), the lidar error is below 2% until a non-dimensional height (z/L) of 0.16 is reached. The error reaches a maximum of 3% at z/L=0.5 and falls below 2% again above z/L= 1.5.* Maybe even set this in context, e.g. with a hill L=1000m and H=100m.

Thank you very much for your suggestion. We have considered this and rephrased the section in the revised version of the manuscript. This discussion is also moved to the newly introduced 'Discussion' section.

**3.4 Influence of surface roughness**

L427: *reference* – which reference?

Please excuse the confusion. What is meant here is the potential flow model, which – as the simplest of the three models – is used as a „reference" or „baseline" simulation (compare lines 221-224). In order to avoid confusion we have decided to always refer to it as the „potential flow model" and have rephrased the sentence in the revised version of the manuscript.

L436: The -2% dashed line is missing in Figure 7 (Right)

As explained above, we have completely removed the -2% dashed lines in the plots in the revised version of the manuscript.

---

## Author Comment (AC2)

**Review: The five main influencing factors on lidar errors in complex terrain, wes-2021-26**

The authors would like to thank the referee very much for the extensive and useful feedback provided and the effort put into this. We appreciate the comments and suggestions very much. They will surely help us to to increase the quality of the manuscript. We find it very valuable to add additional information that is of relevance for the reader.

In the following, we answered all of the referee's questions, comments and suggestions. We state how we use the feedback in the revised version of the manuscript and provide additional information and explaination for the referee whereever needed.

**General comments:**

The authors are discussing one of the critical issues in lidar measurements, namely the error associated with lidar measurements. In particular, this contribution is focusing on the fundamental assumption of flow homogeneity which is the cornerstone of horizontal wind reconstruction techniques. This work split the error into two parts, which are very promising in understanding and deeply analysing the source of errors.

Having said that, this work lacks in many places, especially the discussion section, providing deep insights about the predicted results. Moreover, the newly introduced error classification has been ignored in many of the result subsections. That has been reflected on the conclusion section as well.

Here is a list of general suggestions and points which could improve the manuscript:

- Improve the description of the model setups in order to improve the reproducibility of the work.
- Provide an insightful explanation to the predicted trends in the error.
- Add where it is possible as a ratio of the total error, the two error parts.
- Add where it is possible more than one model results.
- Although turbulence plays an essential rule in atmospheric flow, it is ignore in this analysis. Examining turbulent intensity and/or turbulent kinetic energy could give some insights about the predicted results.
- The current study depends on steady state flow models which quite understandable for the scope and the applications of this work, but the authors could at least discuss this effect. In the end, Lidars do not measure stationary wind conditions.
- Although the error estimation is relative to the model full predicted wind vector, a validation of the model setups will be beneficial. The hill case is well studied in the literature, and there is a great chance to find data for some of the modelled cases in this work.

The specific comments section is reflecting most of the above comments. In general, the manuscript language could be improved in some places, but nothing major.

The citation format is not consistent across the paper, for instance sometime it takes the form of Author (Year) or (Author, year) or (Author (year)). Please follow the journal format and be consistent across the manuscript.

Same applies on the figures, for instance -2% dashed line is missed in some plots.

We would like to thank you very much for your elaborate and helpful general feedback on our manuscript. We are pleased to read that in your oppinion we examine an important topic in the

context of lidar measurement. We are also glad to hear that our approach to separate the two error parts is seen as a promising foundation.

Furthermore, we appreciate your list of suggestions to improve our manuscript. We find it very helpful in reviewing our work and think that it will benefit much from your comments.

Based on your comments and suggestions we have come to the conclusion that it would improve the readability and structure of the manuscript to add a 'Discussion' section that includes many of your suggestions. We have therefore added that section between the 'Methods' and 'Conclusions' sections.  We have moved existing discussions to that section, which makes the results section shorter.

We have extended our discussion according to the following points, which also include your above given feedback:

- Discussion on general findings for all parameters
- Discussion and interpretation of the error parts and their importance, which includes a discussion of the ratio of the error parts, also in dependence of the flow model used and for the different parameterizations
- Discussion of the role of the half-cone opening angle, including references to relevant literature on that topic
- Discussion and interpretation of the influence of roughness, forest and atmospheric stability including explaination of the reason for the influence whereever possible
- Discussion of the implications of the sensitivity of the lidar error estimation to model parameterization and what this means for applications
- Discussion of combinations of different parameters (roughness and stability or forest and stability etc.)
- Considerations regarding the magnitude of the lidar error in comparison to e.g. cup anemometry in complex terrain
- Considerations on the flow models used, including possible limitations and discussion about other models, also including a short discussion about the use of steady-state or transient models
- Discussion on available (and needed) experimental validation of the flow modelling results, including a general discussion about the applicability (and validitaton) of the models for our case

We have included additional literature references where needed to support our discussion. We have also included additonal plots with streamlines from the RANS flow model that help discussing and interpreting the influence of roughness (forest) and atmospheric stabilty on the model results.

We would especially like to thank you four your suggestions regarding the description and discussion of the flow models we use. We have revised the description of the model setups and added more specific information about the meshes, the surface and forest paramterization as well as the resolution. We think the reproducibility has been improved. However, for all models we have also cited relevant literature that describes the models in detail. Additionally, a more detailed model description can be found in the dissertation of the main author.

We have also added additional results plots to show more results from each of the models.

We have reviewed the citations and harmonized them according to the journal rules.

We have removed the -2% line from the plots in the revised version of the manuscript. Instead, we have added a more elaborate discussion about cup anemometer uncertainties in the relevant section.

**Specific comments:**

**0. Abstract:**

The abstract could be improved by including some key results of the study, such as the error increase value due to one or two parameters. Also, adding the conclusion about the error due to the cone angle, since it is one of the main finding of the study and explains the contradictions in the literature.

Thank you for your suggestions. We have slightly changed and extended the abstract. It now includes the following additional information:

Lines 24-25: "Based on the error separation approach it furthermore allows for an in-depth analysis of the influence of reduced half-cone opening angles, explaining contradiction in the previously available literature."

Line 27-28: "The use of a RANS CFD model in conjunction with an appropriate forest model is highly recommended for lidar error estimations in complex terrain, since forest (and roughness) tend to reduce the lidar error."

L11: influencing factor on or of

Thank you, we have corrected this in the revised manuscript.

L14: reformulate to make it more clear

Thank you, we have rephrased the sentence in order to make it more clear what we mean by „complete interpretation".

L15: remove ,eg.,

Thank you, we have corrected this in the revised manuscript.

**1. Introduction:**

L32: Group citation should be in chronological order

Thank you, we have corrected this in the revised manuscript.

L34: I guess you mean the horizontal wind vector/speed reconstruction

Thank you for your comment. We agree that this should be focussed on the horizontal wind vector reconstruction. We have rephrased this in the revised manuscript.

L42: chapter -> section

Thank you for you comment. We agree that „section" suits better and have revised this in the manuscript.

L47: This statement is quite general, the assumption is not valid when there is a significant spatial changes between the measurement points. Therefore within the same complex site, it will depend on the position of the profiler lidar and the cone angle.

Thank you very much for you comment. We agree that – for a given complex terrain site – the lidar error depends on the measurement location (position) and also the cone angle of the lidar. In addition to that, it also depends on other factors that are discussed within the manuscript. However, at this passage in the text, the explanation is meant to be „general". The influence of specific factors is explained in detail in the subsequent text.

For homogeneous flow conditions, e.g. in flat terrain, both, the position of the lidar and its cone angle do not influence measurement accuracy.

We have added a paragraph where we discuss the implications of moving the lidar measurement location from the top of a hill to the flanks or to other type of terrain such as escarpments or valleys in the newly introduced 'Discussion' section.

L49:L51: it is not clear if you are comparing Anotoniou et al. (2007) with Smith et al. (2006), and what is the main contradiction between the two. Please reformulate.

Thank you very much for your comment. We have slightly rephrased this sentence in the revised manuscript, to make it clear that we compare the results from Anoniou et al. (2007) to those from Smith et al. (2006).

L52: This 5-10% error reported by Courtney et al. (2008) is common in complex terrain, please elaborate from that study why? Otherwise it is not clear for the reader why reducing the cone angle from 30deg -> 15deg is necessary.

Thank you very much for your comment.

At the early stage of lidar validation studies in complex terrain a reduction of the distance between the measurement points was proposed. Technically this could be achieved by reducing the lidar half-cone opening angle which brings the measurement points closer together and reduces the volume of the cone that is stretched out by the DBS measurement geometry (compare Figure 2 of the manuscript). The underlying assumption is that a smaller volume of the measurement geometry will reduce the magnitude of spatial changes in the wind speed between the measurement points, which might then also reduce the systematic lidar errors. This is why a reduction of the half cone angle was proposed by Courtney et al. (2008).

However, in later experimental studies (e.g. Bingöl et al. (2009) and Foussekis (2009)) reducing the half cone angle did not reduce the systematic lidar error.

We have rephrased the last sentence of the paragraph in the revised manuscript to make it more precise and to explain the reason for reducing the cone angle.

L59: It is great to mention this contradiction but please elaborate more and mention why there was no significant increase in the error.

Thank you very much for your comment.

We have explained the reason for this in the results and discussion section of the paper in detail, based on the lidar error separation methodology that we developed in our study. The introductory

chapter is meant to give a brief overview over the relevant literature and also to point out gaps that we are trying to fill with our study.

We have decided not to change this section but to elaborate the influence of the half cone angle in more detail in the newly introduced 'Discussion' section. We have also added a back reference to the literature review in the revised manuscript there.

L67: not dependent -> independent . I agree with this conclusion but with larger cone angle, there is a higher change to capture inhomogeneous flow.

Thank you very much for your comment. We agree that „not dependent on" suits better in this context and have rephrased this in the revised manuscript.

Please see our elaborations above regarding the influence of cone angle in the newly introduced 'Discussion' section. Also, we have revised the results and discussions section in order to explain in more detail how the half cone angle influence lidar measurements in complex terrain.

L71: why is it independent of height ?

Thank you for your comment.

The referenced study states that the error is independent of height at the measurement site, since they find no height dependency in the data. There is no further discussion or explanation why that is the case here. In the results and discussion section we generally explain height dependency of lidar errors in complex terrain based on the presented non-dimensional appraoch. However, we agree that a back reference to Foussekis (2009) would be beneficial to discuss the reasing why no dependence on height is found there. We have added that in the newly introduced 'Discussion' section.

From our point of view the explanation is that due to the individual site characteristics the height dependency is too small to be seen in the measurement data.

L86: do you mean RANS CFD here? Please be more specific, CFD is a general term.

Thank you for your comment. Yes, in all studies on the application of CFD tools for the purpose of lidar error estimation, RANS CFD models are used. We have therefore rephrased the relevant passages in the revised manuscript.

L114: Group citation should be in chronological order

Thank you, we have corrected this in the revised manuscript.

L123: Group citation should be in chronological order

Thank you, we have corrected this in the revised manuscript.

L135: Indeed stability has a significant impact on wind profiles and turbulence levels even in flat terrain, but it is not obviously clear to the reader at this point how it could influence the lidar accuracy. Please elaborate more.

Thank you very much for your comment. We have rephrased this in the revised manuscript to make it more clear to the reader why atmospheric stability does have an influence on lidar errors in complex terrain.

We expect atmospheric stability to change the flow patterns above complex terrain. Changed flow patterns will then affect the lidar errors, which makes the lidar error indirectly dependent on atmospheric stability. We support this with relevant literature on flow over complex terrain in neutral and stratified atmospheric conditions (e.g. Ross, A. N., Arnold, S., Vosper, S. B., Mobbs, S. D., Dixon, N., and Robins, A. G.: A comparison of wind-tunnel experiments and numerical simulations of neutral and stratified flow over a hill, Boundary-Layer Meteorol, 113, 427–459, https://doi.org/10.1007/s10546-004-0490-z, 2004.).

We have added this to the newly introduced 'Discussion' section. To explain better, how flow patterns change due to changes in atmospheric stability, we have added a plot with streamlines over a hill from our simulations.

L137: is another factor -> do you mean 'are another factors' ?

Thank you for your comment. No, we mean that the half-cone opening angle must also be considered as an influencing factor on lidar errors in complex terrain. We agree that the sentence is a bit hard to read and have rephrased it in the revised manuscript to make it more clear.

L145: Group citation should be in chronological order

Thank you, we have corrected this in the revised manuscript.

**2. Methods**

I suggest to have some subsections here, at least two, one for the model and one for the lidar error.

We agree with your suggestion and have split up the Methods section into three subsections in the revised version of the manuscript:

2.1 General considerations
2.2 Flow modelling methods
2.3 Lidar error correction

We think that we have increased readability of the Methods section by this.

L178: forest parameters are not necessarily surface parameters, for instance a typical forest model parameters have to be mapped as volume values.

We agree with your comment that forest model parameters are mapped as volume values. This is of course also the case in the RANS CFD model we use (Meteodyn WT). We have changed „surface parameters" to „model parameters" in the revised manuscript, which we think is a more general term.

L191: Group citation should be in chronological order

We have corrected this in the revised manuscript.

L196: Isolated one sentence should not be a separate paragraph. Please reformulate it.

Thank you, we have rephrased this in the revised manuscript.

L200: The literature is full of similar cases for Gaussian and Cosine hills and the similarity parameters are well established. Please give a background about this case and cite the appropriate literature. Also add the mathematical formula of the hill or cite the source.

Thank you very much for your suggestions. We have added the formula for the Gaussian hill with an appropriate literature reference (Feng, J. and Shen, W. Z.: Wind farm layout optimization in complex terrain: A preliminary study on a Gaussian hill, J. Phys.: Conf. Ser., 524, 12146, https://doi.org/10.1088/1742-6596/524/1/012146, 2014.).

We have also added a reference to a recent review on boundary-layer flow over complex topography, which provides an elaborate overview about the relevant literature (Finnigan et al. 2020, https://doi.org/10.1007/s10546-020-00564-3).

L200: H and L should be formatted as math for consistency.

Thank you, we have corrected this in the revised manuscript.

L222: win flow → wind flow

Thank you, we have corrected this in the revised manuscript.

L240: The used models are steady, aren't they? Add this to the description of the models

Thank you for your comment. Yes, that is correct; we have added this information in the revised manuscript to complete the model description.

L241: Add the forest parameters, (low, medium and high) are very general terms

Thank you very much for your comment. We agree that we should be more precise on these values. Forest density in Meteodyn WT is defined by the drag force coefficient $C_d$. We have added the values for the three settings low, medium and high to the revised manuscript. However, we have not explained the forest model in detail. This is done in the literature we cite and also in the dissertation of the main author.

L241: is the forest applied uniformly above the ground? Or limited to the hill. Please elaborate more on the forest model setup.

Yes, the forest (like the roughness parameters) is applied uniformly above the ground for the whole model domain. We have added more details on the forest model setup in the revised version of the manuscript and also added this information.

L247: This was done for a low roughness case, ……. It is not clear what do you mean here.

Please excuse the confusion. We have rephrased this paragraph in the revised version of the manuscript to make it easier to understand. What we mean is that we did not change atmospheric stability for all different surface and forest parameterizations but only for three selected cases.

L249: exemplaryly ?

Thank you, we have corrected this in the revised manuscript.

L250: The model description lack any information about the used meshes. For instance, in addition to the mesh horizontal resolution and domain size, the first cell size is an important parameter especially with high roughness values.

Thank you very much for your comment.

We have added more information about the used mesh resolutions in the model description. For all models and all parameter variations the same mesh settings are used. Horizontal resolution is always <= 10m in the proximity of the lidar location. Because of the individual properties of the models, we use slightly different resolutions for each of them. Resolution in the proximity of the lidar location and the hills is always constant and comparable between all models. We ensured that horizontal and vertical resolution is small enough to resolve the wind flow evolution between the measurement points of the lidar.

L255: One could start a subsection here.

Thank you. We agree with this suggestion and have split up the section here (see comment above).

L286: chapter → section

We agree and have corrected this in the revised manuscript.

L303: Eq(6), is this assumption applied in this work?

Thank you very much for your comment.

This assumption is not applied here but it is given for the sake of completeness. Generally, equation (5) is used. However, for the potential flow model equation (6) is valid since it is a symmetric model where inflow and outflow angles are equivalent.

(!) Please note that we have introduced an additional equation in the revised manuscript – the Gaussian hill definition – that is now equation (1). Following equation numbers have therefore been shifted by 1. Above given equation numbers refer to the original manuscript.

L311: equation 6 → equation 5?

Please excuse the error. We have corrected this in the revised manuscript.

L313: Are equation (9) and (1) equivalent? In other terms, if one estimated the error based on Eq(1) and Eq(9) will get the same value or not? Have the authors tested that?

Thank you very much for your question.

Actually, the two equations are not equivalent. However, we have tested the deviations between equations (1) and (9) before and they are very small (neglegible for most hills). Equation (9) is a profound approximation of both, the flow curvature and speed-up induced lidar errors.

There are two main assumption that are used when deriving the equations:

1.  The ratio $k$ between outflow and inflow wind speed ($k = V_{out}/V_{in}$) is set to 1, assuming that both wind speeds are the same. This removes $V_{out}$ and $V_{in}$ from the equation.

2. The horizontal wind speed at the lidar location $u_L$ is defined as the mean of $u_{in}$ and $u_{out}$. This makes it possible to cancel $u_L$ out of the equation, when inserting equation 4 into the lidar error definition equation 1 (left part).

These two assumptions enable us to derive an equation for the flow curvature induced lidar error $\varepsilon_c$ that is only dependent on inflow angle $\alpha$, outflow angle $\beta$ and half-cone opening angle $\varphi$.

With respect to these assumptions, we have slightly changed the resulting equations (5) and (6) and replaced the equality sign by an approximately equal sign. This also underlines the fact that equations (1) and (9) are not equivalent. The assumptions above are mentioned in the text.

(!) Please note that we have introduced an additional equation in the revised manuscript – the Gaussian hill definition – that is now equation (1). Following equation numbers have therefore been shifted by 1. Above given equation numbers refer to the original manuscript.

L313: Is the same approach applicable om three-dimensional flow? If so, it would be beneficial to end this part with the general formulation.

Thank you very much for your comment and your suggestion.

The same approach should also be applicable to three-dimensional flow. However, it will also become more complex since deviations in wind direction, i.e. changes in both, the $u$ and $v$ component of the wind vector will probably become relevant. An illustration for this is included in the dissertation of the main author (Figure 3.6 in Klaas, T.: Model-based study of the five main influencing factors on the wind speed error of lidars in complex and forested terrain, Universitäts- und Stadtbibliothek Köln, Köln, Online-Ressource, 2020.).

On the other hand, the two-dimensional approach is a very good approximation for the three dimensional case. This is because the radial wind speeds measured by the lidar are projections of the three-dimensional wind vector to the line-of-sight of the laser beam at the measurement locations. For wind directions parallel to two of the lidar beams (as in the two-dimensional simplification), the radial wind speeds on the lidar beams perpendicular to the wind direction will be zero. For wind directions inbetween, it will be split up to the lidar beams.

We have decided not to add a three-dimenional formlation, because we do not see much benefit from it for our study. The advantage of sticking to the two-dimensional formulation is its simplicity in interpretation. In real-life and 3D applications we suggest to directly use the general formulation which automatically takes into account all error parts without assumptions and simplifications.

**3. Results and discussion**

L323: exemplaryly →exemplary?

Thank you, we have corrected this in the revised manuscript.

**Subsection 3.1:**

Fig 4: for the right hand side figures a, b, c, it is more reasonable (and more common for such case) to use normalized x, and y axis with respect to L or H. Also the choice of y-coordinate is confusing since the vertical component is considered z-dir in the rest or the paper.

Thank you very much for your comment. We have changed the y-axis caption to „z-coordinate", which is now consistent with the rest of the manuscript.

Regarding the scaling of the right hand side of the figure, we agree that it would be more common to use normalized axes. However, since the difference in size of the hills, especially on the x-coordinate, is very large, this would result in very mall representations of the cases a) and b). Normalized axes would therefore heavily decrease readability of the plot. We think that the key message of the figures, i.e. the size relation between the lidar measurement geometry, the hill size and the position of the measurement points with regards to the hill shape, is obvious also without normalized axes. We would therefore like to keep the axes as they are in the revised version.

L378: speep-up → speed-up

Thank you very much, we have corrected this in the revised manuscript.

L378: what about the error variation due to the model? Each model has a different error prediction and it would be good to show it for this case.

Thank you very much for your comment.

We agree that it would enrich the discussion, if we include results from the other two model (WasP Engineering and Meteodyn WT). We left them out before because we want to keep the number of figures and the length of the manuscript reasonable short. Detailed and complete results can be found in the dissertation of the main author.

Nevertheless, we have added two additional result plots in the revised version of the manuscript that show results for those models for each of the four $H/L$ ratios. We have also added description and discussion to the manuscript text. We agree with you, that it will then be easier and more straightforward to discuss the influence of roughness in the following subsections.

**Subsection 3.2:**

L380: Add to the text the used model, it is not enough to mention it in the figure caption only.

Thank you for your suggestions. We have added this in the revised manuscript.

L380: I think this subsection is essential for this study because it shows the two error parts. So it is essential to show the effect of the model on the error distribution. For example, the speed-up error ratio (with respect to the total) should be relatively higher in RANS model compared to potential flow model. Showing the ratio of each part across the three models will enrich the discussion.

Thank you very much for your comments.

We are happy to hear that especially the separation of the total error into its two parts is getting so much positive feedback. Detailed results on this can be found in the dissertation of the main author.

Considering your feedback, we have decided to include a paragraph that discusses the error parts for the two other models in the newly introduced 'Discussion' section. We have thought about adding more result figures, which show the different error parts or ratios for the different model (and for different parameterizations). However, we have come to the descision that this will lengthen the manuscript too much. We would therefore like to keep the already included figures and discuss the other models in the text.

L395: is the predicted flow of all cases and models fully attached? Although it could happened downstream directly above the ground, it may influence the measurement heights.

Thank you very much for your comment.

When increasing roughness and especially when introducing forest into the RANS CFD model, we can see that close to the ground, respectively inside the forest, that we tend to have high turbulence and also flow seperation. These effects cause the flow field to change up to the measurement heights of the lidar and are one of the main reasons for the resulting, decreased lidar errors. Enhanced flow separation in forested cases is also discussed in literature and we have added relevant literature to the discussion (e.g. Belcher, S. E., Finnigan, J. J., and Harman, I. N.: Flow through forest canopies in complex terrain, Ecological Applications, 1436–1453, 2008.).

We have included this in the newly introduced 'Discussion' section in the revised manuscript.

**Subsection 3.3**

I have similar comments as above, why it is limited to one model here. Especially, potential flow is symmetric and does not show the expected diverse behavior in the curvature and speed-up errors.

Thank you very much for your comment.

The aim of subsection 3.3 is to illustrate the influence of the cone opening angle on both error parts and especially explain the opposed effects on the total lidar error. This is done best based on the results from the simple potential flow model.

We agree with your comment. The more sophisticated models are no longer symmetric and will show diverse behavior in the error parts. We have modelled / tested this for some cases and have found examples where the half-cone opening angle might be an important parameter that increases or decreases the total lidar error.

We have decided to add one exemplary result plot based on results from Meteodyn WT with increased roughness length that illustrates such a case. We have also added a discussion on the in the newly introduced 'Discussion' section.

**Subsection 3.4**

L425: as the above mentioned comments, without including the other models in comparison to the otential flow, it would be hard to isolate the influence of the roughness. The increase in the max error here is not because of the roughness only.

Thank you very much for your comments.

Section 3.4 inlcudes all three models. Both, WAsP Engineering and Meteodyn WT are shown in comparison to the potential flow model, each for three different roughness length. Because we do not want to show too many plots – with roughly the same result – we are limiting to an $H/L$ ratio of 0.3. We think that in this way, we are able to isolate the influence of roughness on the total lidar error quite good for both models.

We agree that of course different models show different results that are not caused by terrain roughness. Because of that and following your comment regarding Subsection 3.1 we have added results for low roughness ($z\_0 = 0.005m$) for all $H/L$ ratios for the other two models (WasP Engineering and Meteodyn WT) in Subsection 3.1 as well. By that we can relate to the low roughness cases in Subsection 3.1 when discussing the results for higher roughnesses in Subsection 3.4.

We have included this discussion in the newly introduced 'Discussion' section.

L436: This subsection lacks any explanation for why the error decrease which increasing the roughness values. Probably it is related to turbulence mixing.

Thank you for your comment.

As already described above we have now introduced a 'Discussion' section. There we are also discussing the influence of roughness on flow over a hill and reference to relevant literature. We also have included an additional Figure that shows the streamlines above a hill from our simulation for different roughness length. With this figure, it is easier to discuss and explain the implications of increased roughness on the flow field.

L436: it would be interesting to see the influence of roughness on the ratio of the two errors.

Thank you very much for your comments.

As already stated in our above comment: Considering your feedback, we have decided to include a paragraph that discusses the error parts for the two other models in the newly introduced 'Discussion' section. We have thought about adding more result figures, which show the different error parts or ratios for the different model (and for different parameterizations). However, we have come to the descision that this will lengthen the manuscript too much. We would therefore like to keep the already included figures and discuss the other models in the text.

**Subsection 3.5**

As mentioned in my comments on the method section, the forest parameters are not clear. Did you use the same LAD or LAI?

Thank you very much for you comment. No, we have not used LAD or LAI data / parameterizations. Forest density in the RANS CFD model Meteodyn WT is considered by different $C_d$ values. We have added this to the description of the model in the text.

L445:446: epsilon is not math format

Thank you, we have corrected this in the revised manuscript.

Fig8: I do not see the point of including the forest and stability in the same figure.

Thank you very much for your comment. The intention of including both results in one figure was to keep the number of figures low. However, following your comment this is obviously more confusing than helpful. We have therefore decided to split the two result plots into two figures in the revised manuscript.

Fig8: there is no explanation for this behavior, why the error decrease with increasing the forest height, or why both 20m and 30m cases have the same max error.

Thank you very much for your comment.

As already described above we have now introduced a 'Discussion' section. There we are also discussing the influence of forest on flow over a hill and reference to relevant literature. Enhanced flow separation in forested cases is also discussed in literature and we have added relevant literature

to the discussion (e.g. Belcher, S. E., Finnigan, J. J., and Harman, I. N.: Flow through forest canopies in complex terrain, Ecological Applications, 1436–1453, 2008.).

We have looked into the results for the 20m and 30m cases for forested terrain. The influence of an increase from 20 to 30 m height is small. However, the errors are not the same but slightly different for all $z/L$ ratios. We have added a more elaborate discussion on this to the newly introduced 'Discussion' section.

L451: again this section focuses on the total error and does not show the two error parts which are the main promise of this work.

Thank you very much for your comments.

As already stated in our above comment: Considering your feedback we have decided include a paragraph that discusses the error parts for the two other models in the newly introduced 'Discussion' section. We have thought about adding more result figures, which show the different error parts or ratios for the different model (and for different parameterizations). However, we have come to the descision that this will lengthen the manuscript too much. We would therefore like to keep the already included figures and discuss the other models in the text.

**Subsection 3.6**

It is not mentioned what are the model parameters behind these stability classes. For instance, Table 3 includes three stable conditions, which on is included here?

Thank you very much for you comment.

We agree that it remains unclear which of the ‚stable' classes is shown in Figure 8. However, four selected stability classes are shown that cover the whole spectrum of stability classes available in the model. Starting with ‚very unstable' (class 0), then neutral (class 2) as well as stable (class 6) and ‚strongly stable' (class 9) (compare Table 3). We have revised the figure legend and included the class numbers. Since results for other stability classes are not shown in the manuscript, we have decided to remove Table 3 and rephrase the relevant paragraph that now explains the stability classes used.

Same comments as before, what is the effect of thermal stratification on the two errors? Any insights why does the error decrease with moving towards stable condition?

Thank you very much for your comment.

We expect atmospheric stability to change the flow patterns above complex terrain. Changed flow patterns will then affect the lidar errors, which makes the lidar error indirectly dependent on atmospheric stability. We support this with relevant literature on flow over complex terrain in neutral and stratified atmospheric conditions (e.g. Ross, A. N., Arnold, S., Vosper, S. B., Mobbs, S. D., Dixon, N., and Robins, A. G.: A comparison of wind-tunnel experiments and numerical simulations of neutral and stratified flow over a hill, Boundary-Layer Meteorol, 113, 427–459, https://doi.org/10.1007/s10546-004-0490-z, 2004.).

We have added this to the newly introduced 'Discussion' section. To explain better, how flow patterns change due to changes in atmospheric stability, we have added a plot with streamlines over a hill from our simulations.

**4. Conclusion**

It is highly advised to revisit this section once applying the above comments. It could benefit also from adding more insights about this future work, it could be a subsection or one paragraph.

Thank you very much for you comment.

We have rephrased the conclusions section in the revised version of the manuscript. We have especially considered the referee's comments and suggestions in the revised verison. We have also added a short paragraph on (possible) future work.

---

## Referee Report (RR1)

Review of: Kaas and Emeis, The five main influencing factors on lidar errors in complex terrain (revised).

General

The revised version of the paper is excellent. Both the structure and the language have been lifted to a level that do justice to the technical content. The addition of the discussion section has in particular added the insight that was missing from the previous version.

I have only a couple of smaller suggestions (and these should be the last!):

- The authors have incorporated my comments concerning the (lack of) accuracy of cup anemometers in complex terrain. I would suggest that they go one step further and give us some numbers since this really puts all the lidar/mast comparisons in a somewhat different light. E.g the Windsensor cup anemometer (high quality and widely used) has class numbers of 1.32A and 3.71B. A typical standard uncertainty (68% confidence level) on a mast would therefore be 1.4% @ 10 m/s in flat terrain (class A) and 2.4% @ 10 m/s in complex terrain class B. Double these numbers for the more typically used 95% confidence limit!
- The development of section 2.3 Lidar error correction could be even better (but I do understand it now). In particular I would suggest that the authors develop firstly the equations for the ideal homogeneous case (same flow at both sensing positions). Actually this equation first appears in a box in figure 3. I think it would help the general understanding to develop this first then demonstrate how this gives the wrong answer when the flow is no longer truly homogeneous.

More detailed comments follow:

P2, L40  (suggestion) 'to test lidars at these sites to assess their applicability' -> 'to examine the accuracy of lidars at such sites'

P3, L93 '…FCR over-corrects for the lidar error' – its probably within the measurement uncertainty.

P6, L183-189. Are you suggesting using the raw lidar data to check the flow model that you will then use to correct the lidar? Maybe this is getting dangerously circular??

P10, L283   'The flow inclination'…….'therefore introduces an error component' – this reads as if it's the flow inclination itself that gives the error. I'm sure the authors don't believe this! Please re-word so that its clear it's the change in inclination angle – here developing the homogeneous case first would help the understanding (my main comment #2).

P11 Figure 3 – see my main comment #2. The equation in the box just pops out of nowhere!

P12, l313 – you need $\beta = -\alpha$, not $\alpha = \beta$! Else the numerator of eq(6) becomes zero!

P12 generally – maybe just take the symmetric case for $\varepsilon_s$ as well (you get $\Delta u / u_L$ if I understood this correctly).

P21 Discussion – maybe one important point that you miss here is that both error components have the same sign (is this ever not true?).

---

## Referee Report (RR2)

Review: The five main influencing factors on lidar errors in complex terrain, wes-2021-26

The manuscript has improved considerably compared to the initial version in many aspects. The authors responded to all comments and updated the article accordingly. I am glad to see the paper on this level now. However, there are few points are overlooked or may be were not very clear in my first review:

General comments:

- Although turbulence plays an essential rule in atmospheric flow, it is ignore in this analysis. Examining turbulent intensity and/or turbulent kinetic energy could give some insights about the predicted results.

    - The turbulence level which is influenced with the five factors investigated here could shed some light on the difference between different cases. The speed-up factor could be very sensitive to the turbulence intensity.

- The current study depends on steady state flow models which quite understandable for the scope and the applications of this work, but the authors could at least discuss this effect. In the end, Lidars do not measure stationary wind conditions.

    - This point hasn't addressed. The highest fidelity model used in this work is steady RANS which is for stationary isotropic turbulence which is far off the wind conditions measured by the Lidar. The point here, if one predict the error as proposed, has to be very careful when applying the correction.

Specific comments:

L380: I think this subsection is essential for this study because it shows the two error parts. So it is essential to show the effect of the model on the error distribution. For example, the speed-up error ratio (with respect to the total) should be relatively higher in RANS model compared to potential flow model. Showing the ratio of each part across the three models will enrich the discussion.

Thank you very much for your comments.
We are happy to hear that especially the separation of the total error into its two parts is getting so much positive feedback. Detailed results on this can be found in the dissertation of the main author. Considering your feedback, we have decided to include a paragraph that discusses the error parts for the two other models in the newly introduced 'Discussion' section. We have thought about adding more result figures, which show the different error parts or ratios for the different model (and for different parameterizations). However, we have come to the descision that this will lengthen the manuscript too much. We would therefore like to keep the already included figures and discuss the other models in the text.

Regarding showing the two parts of error which was mentioned more than once in the first review round, I agree with the author that it would be too much to show it for all models for the five parameters. I find Figure 7 very interesting, it shows the different behaviour of the two parts and the order of each one. I think it would be a missed opportunity to not show a similar plot for the other factors.

L178 models repeated

L240 A constant horizontal resolution of 10 m was used in the proximity of the lidar location and the minimum resolution is 25 m.

- It is not clear what that does mean

L555 forest parameter influence on error

- This conclusion isn't very clear. From modelling perspective, there is no doubt that if the site has forested area, it must be included in the model with the right parameters. In Klaas (2015), since measurements were available, it was clear the right forest parameter reduce the gap between the model and the measured data which is expected. This work shows that increasing forest height reduce the error which a good point stress on. Typically a forsted site is considered to be complex and this study finding states that forest does not add error compared to less forested site.  That make the L27(abstract) and here kind of misleading to some readers.

---

## Author Response (AR2)

**Review of: Kaas and Emeis, The five main influencing factors on lidar errors in complex terrain (revised).**

The authors would like to thank the referee very much again for reading the revised version of the manuscript and providing additional, valuable feedback and suggestion on how to further improve it.

In the following, we have answered to all of the referee's questions, comments and suggestions. We also state how we use the feedback in the revised version of the manuscript.

**General**

The revised version of the paper is excellent. Both the structure and the language have been lifted to a level that do justice to the technical content. The addition of the discussion section has in particular added the insight that was missing from the previous version.

I have only a couple of smaller suggestions (and these should be the last!):
- The authors have incorporated my comments concerning the (lack of) accuracy of cup anemometers in complex terrain. I would suggest that they go one step further and give us some numbers since this really puts all the lidar/mast comparisons in a somewhat different light. E.g the Windsensor cup anemometer (high quality and widely used) has class numbers of 1.32A and 3.71B. A typical standard uncertainty (68% confidence level) on a mast would therefore be 1.4% @ 10 m/s in flat terrain (class A) and 2.4% @ 10 m/s in complex terrain class B. Double these numbers for the more typically used 95% confidence limit!
- The development of section 2.3 Lidar error correction could be even better (but I do understand it now). In particular I would suggest that the authors develop firstly the equations for the ideal homogeneous case (same flow at both sensing positions). Actually this equation first appears in a box in figure 3. I think it would help the general understanding to develop this first then demonstrate how this gives the wrong answer when the flow is no longer truly homogeneous.

Thank you very much for you general feedback on the revised version of our manuscript and for the effort you put in reading this version. We are glad to hear that we have mostly met your expectations based on your comments and feedback on the first version.

- We understand your wish to add more detailed information on the uncertainty of cup anemometers in flat and complex terrain. We have therefore added an example with classification numbers for a Thies First Class cup anemometer from the literature. We have also calculated the standard uncertainty including calibration and mounting uncertainties.
- We have extended the development of our correction equations and added that flat terrain case. We agree that this will help the reader to follow and understand the situation in complex terrain.

**More detailed comments follow:**

P2, L40 (suggestion) 'to test lidars at these sites to assess their applicability' -> 'to examine the accuracy of lidars at such sites'

Thank you, we have changed the sentence according to your suggestion.

P3, L93 '…FCR over-corrects for the lidar error' – its probably within the measurement uncertainty.

Thank you. We have slightly revised this paragraph. We think it becomes clear now that the fact that FCR tends to over-correct in complex terrain has been found in several studies. However, we agree that in the mentioned case, the over-correction could be within the uncertainty margins of the measurement and have added this to the text.

P6, L183-189. Are you suggesting using the raw lidar data to check the flow model that you will then use to correct the lidar? Maybe this is getting dangerously circular??

Thank you for your comment. We agree that the paragraph reads a bit misleading and have therefore slightly rephrased it. We also agree that it is a challenge to avoid circularity when using the lidar data for model validation and then the model for lidar error correction. We have proposed to use additional data for model validation wherever possible before doing the lidar correction in the revised version to make it clearer.

P10, L283 'The flow inclination'…….'therefore introduces an error component' – this reads as if it's the flow inclination itself that gives the error. I'm sure the authors don't believe this! Please re-word so that its clear it's the change in inclination angle – here developing the homogeneous case first would help the understanding (my main comment #2).

Thank you for your comment. We agree that this reads misleading and have slightly rephrased the paragraph in the new version.

P11 Figure 3 – see my main comment #2. The equation in the box just pops out of nowhere!

Thank you. As already stated above, we have extended the development of our equations and the mentioned equation is now part of the text.

P12, l313 – you need $\beta=-\alpha$, not $\alpha=\beta$! Else the numerator of eq(6) becomes zero!

Thank you, we have corrected this in the revised version.

P12 generally – maybe just take the symmetric case for $\varepsilon_s$ as well (you get $\Delta u/u_L$ if I understood this correctly).

Thank you, we have added the symmetric case in the revised version to make it more complete.

P21 Discussion – maybe one important point that you miss here is that both error components have the same sign (is this ever not true?).

Thank you for your comment. This should always be true and we agree that we have missed this information in the discussion. We have added a short sentence to clarify this fact.

**Review: The five main influencing factors on lidar errors in complex terrain, wes-2021-26**

The authors would like to thank the referee very much again for reading the revised version of the manuscript. We thank you for providing us additional, valuable feedback and suggestion on how to further improve it.

In the following, we have answered to all of the referee's questions, comments and suggestions. We also state how we use the feedback in the revised version of the manuscript.

The manuscript has improved considerably compared to the initial version in many aspects. The authors responded to all comments and updated the article accordingly. I am glad to see the paper on this level now. However, there are few points are overlooked or may be were not very clear in my first review:

**General comments:**

- Although turbulence plays an essential rule in atmospheric flow, it is ignore in this analysis. Examining turbulent intensity and/or turbulent kinetic energy could give some insights about the predicted results.
    - The turbulence level which is influenced with the five factors investigated here could shed some light on the difference between different cases. The speed-up factor could be very sensitive to the turbulence intensity.
- The current study depends on steady state flow models which quite understandable for the scope and the applications of this work, but the authors could at least discuss this effect. In the end, Lidars do not measure stationary wind conditions.
    - This point hasn't addressed. The highest fidelity model used in this work is steady RANS which is for stationary isotropic turbulence which is far off the wind conditions measured by the Lidar. The point here, if one predict the error as proposed, has to be very careful when applying the correction.

Thank you very much for your general feedback and comments. We understand your point regarding the effects of turbulence and instationarity on the actual lidar measurements. We also agree that the five factors we have investigated significantly influence the turbulence in the wind flow.

In the end of the conclusions and outlook section we have briefly discussed the possibility to use more advanced, i.e. non-stationary flow models in order to better explain the scattered results of lidar measurement errors in 10-minute statistics. With such a model we could also analyze instationary effects such as recirculation of detached flow. However, the correction factors resulting from the three models used in this study are meant to be applied to 10-minute statistics time series. They will mainly help to reduce the mean errors and not the error of a single data point.

Analyzing turbulence for the three different models (and for more advanced models) would be an interesting task and worth another, separate study. However, we would rather like to keep our analysis close to the model parameters that are available to the model user. This will help the reader to better understand how to parameterize the models for a good lidar error estimation. We have discussed the effects of roughness elements and forest on turbulence and the flow field with regards to the relevant literature in line 125. From there on it would be possible to study the effects of the different influencing factors on turbulence as well for the interested reader. We hope that we could answer your comments to your satisfaction although we have decided not to include more detailed results on turbulence or turbulent kinetic energy in the model results.

**Specific comments:**

L380: I think this subsection is essential for this study because it shows the two error parts. So it is essential to show the effect of the model on the error distribution. For example, the speed-up error ratio (with respect to the total) should be relatively higher in RANS model compared to potential flow model. Showing the ratio of each part across the three models will enrich the discussion.
Thank you very much for your comments.

*(We are happy to hear that especially the separation of the total error into its two parts is getting so much positive feedback. Detailed results on this can be found in the dissertation of the main author. Considering your feedback, we have decided to include a paragraph that discusses the error parts for the two other models in the newly introduced 'Discussion' section. We have thought about adding more result figures, which show the different error parts or ratios for the different model (and for different parameterizations). However, we have come to the descision that this will lengthen the manuscript too much. We would therefore like to keep the already included figures and discuss the other models in the text.)*

Regarding showing the two parts of error which was mentioned more than once in the first review round, I agree with the author that it would be too much to show it for all models for the five parameters. I find Figure 7 very interesting, it shows the different behaviour of the two parts and the order of each one. I think it would be a missed opportunity to not show a similar plot for the other factors.

Thank you very much for your comment. We agree that it would be interesting to show even more and detailed results on this. However, we still think that this would extend the content of the study too much and would like to stick to the already included figures and results.

L178 models repeated

Thank you, we have corrected this in the revised version.

L240 A constant horizontal resolution of 10 m was used in the proximity of the lidar location and the minimum resolution is 25 m - It is not clear what that does mean

Thank you, we have rephrased the section in order to improve the description of the mesh resolution. What we meant is that we have placed a so-called 'mapping' area in the inner region of the mesh around the lidar location. The mapping has a constant resolution of 10 m. Then the grid is becomes less resolved towards the outer boundary until a minimum resolution of 25 m is reached.

L555 forest parameter influence on error - This conclusion isn't very clear. From modelling perspective, there is no doubt that if the site has forested area, it must be included in the model with the right parameters. In Klaas (2015), since measurements were available, it was clear the right forest parameter reduce the gap between the model and the measured data which is expected. This work shows that increasing forest height reduce the error which a good point stress on. Typically a forsted site is considered to be complex and this study finding states that forest does not add error compared to less forested site. That make the L27(abstract) and here kind of misleading to some readers.

Thank you very much for your comments and suggestions. We understand your concerns and read the mentioned section again. However, we think that in the abstract it becomes clear that forest on the one hand contributes to the complexity of the site and the flow. On the other hand, this increased complexity – surprisingly – leads to reduced lidar errors. This in counterintuitive because – as your mention – forest is considered to increase the complexity of a given site. We have added this to the discussions section to underline this finding.